# FINERS: Fine-grained Reasoning and Segmentation of Small Objects with Reinforcement Learning

**Lu Zhang[1], Jiazuo Yu[1], Haomiao Xiong[1], Ping Hu[2], Yunzhi Zhuge[1],\*Huchuan Lu[1], You He[3]**

[1]Dalian University of Technology, Dalian, China
[2]University of Electronic Science and Technology of China, Chengdu, China
[3]Tsinghua Shenzhen International Graduate School, Shenzhen, China
zhangluu@dlut.edu.cn, yujiazuo@mail.dlut.edu.cn
https://iiau-zhanglu.github.io/FINERS/

## Abstract

Multi-modal Large Language Models (MLLMs) have shown remarkable capabilities across a wide range of vision-language tasks. However, due to the restricted input resolutions, MLLMs face significant challenges in precisely understanding and localizing visual details in high-resolution images—particularly when dealing with extra-small objects embedded in cluttered contexts. To address this issue, we propose FINERS, a two-stage MLLM-based reinforcement learning framework for jointly reasoning and segmenting extremely small objects within high-resolution scenes. FINERS adopts a coarse-to-fine pipeline comprising Global Semantic Exploration (GSE) and Localized Perceptual Refinement (LPR). Specifically, GSE performs instruction-guided reasoning to generate a textural response and a coarse target region, while LPR refines this region to produce an accurate bounding box and segmentation mask. To couple the two stages, we introduce a locate-informed retrospective reward, where LPR's outputs are used to optimize GSE for more robust coarse region exploration. Additionally, we present FINERS-4k, a new dataset for evaluating MLLMs on attribute-level reasoning and pixel-level segmentation on subtle, small-scale targets in complex high-resolution scenes. Experimental results on FINERS-4k and public datasets demonstrate that our method consistently outperforms state-of-the-art MLLM-based approaches on both instruction-guided segmentation and visual reasoning tasks.

## 1 Introduction

Recently, Multi-modal Large Language Models (MLLMs) [1, 2, 3, 4] have achieved remarkable success in a variety of vision-language tasks, such as visual question answering, referring expression comprehension, and instruction-guided segmentation. Among these tasks, one foundational challenge is instruction-guided reasoning and segmentation — a capacity of not only understanding what the user is asking, but also where in the image the referred object appears at the pixel level. Some early attempts [3] integrate MLLMs [4, 5, 6] with foundational segmentation models [7, 8], enabling joint language generation and object segmentation for more interactive and interpretable visual understanding. However, these methods are tailored to standard-resolution images and large, prominent objects, where spatial structures are easily accessible to the model's visual backbone. Their heavy reliance on global visual-semantic alignment becomes increasingly unreliable in scenes with dense layouts and small, low-saliency targets (see Fig. 1 (a)).

To address this challenge, the research community has begun to explore the capacity of MLLMs to perceive fine-grained, detailed objects within high-resolution images. To mitigate detail degradation

---

*Corresponding authors

39th Conference on Neural Information Processing Systems (NeurIPS 2025).

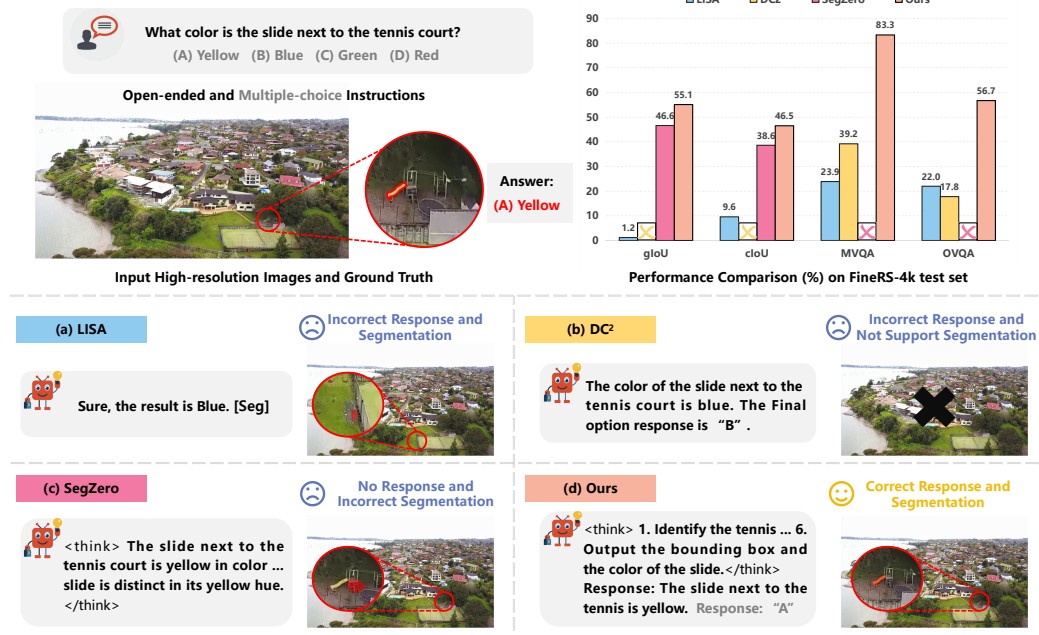

Figure 1: Given a user instruction and a high-resolution image, existing MLLMs face limitations in effectively reasoning and segmenting small objects. (a) MLLMs (e.g., LISA [3]) designed for standard-resolution images fail to generalize to small objects. (b) MLLMs (e.g., DC$^2$ [10]) developed for high-resolution images only support optional question answering and lack localization ability. (c) MLLMs (e.g., SegZero [16]) with RFT fail to produce explicit answers and accurate segmentation masks within a unified framework. (d) Our FINERS combines a coarse-to-fine perception pipeline with reinforcement learning, enabling unified, precise instruction-guided reasoning and segmentation of small objects in high-resolution images.

caused by image downsampling, existing methods [1, 9, 10] mimic human visual perception by decomposing high-resolution images into smaller patches to achieve local vision-text alignment. However, due to the scarcity of high-resolution data, these methods typically adopt a training-free pipeline, where the absence of supervised fine-tuning limits their perception accuracy in complex scenarios (as shown in Fig. 1 (b)). More importantly, the lack of precise localization ability restricts their scalability and applicability to downstream tasks demanding pixel-level grounding and spatial reasoning.

Recent studies [11, 12, 13] have revealed that LLMs can generalize effectively to domain-specific tasks with only thousands of training samples. Moreover, incorporating a "thinking" process prior to answering can significantly enhance their reasoning ability. The core technique behind this improvement is Reinforcement Fine-Tuning (RFT) [12, 14, 13], which enables LLMs to be emergently optimized for downstream tasks via data-efficient fine-tuning. The success of RFT has driven the extension of vision RFT [14, 15, 16] to empower MLLMs across a variety of vision-language tasks, including image classification [14], object detection [14], and reasoning segmentation [16]. However, due to input resolution limitations, these methods [16] still struggle to capture fine-grained details, and cannot simultaneously generate explicit answers and segmentation masks without a unified multi-task reward mechanism (see Fig. 1 (c)).

To address the above issues, we propose FINERS, a two-stage MLLM framework for instruction-guided reasoning and segmentation of small objects in high-resolution images. FINERS is designed to jointly optimize semantic reasoning and spatial perception through a coarse-to-fine pipeline. Specifically, a Global Semantic Exploration (GSE) module first performs instruction-guided reasoning to produce both a textual response and a coarse target region that contains the referred objects inside. The subsequent Localized Perceptual Refinement (LPR) module then refines this region by generating an accurate bounding box and the corresponding segmentation mask. To reduce reliance on heavy supervised fine-tuning, we integrate vision RFT into our framework and design effective rewards to simultaneously address object reasoning and segmentation. Specifically, in addition to

basic region/box regularization rewards, we introduce a response reward to encourage the model to simultaneously generate both textual answers and object boxes. Furthermore, to effectively couple these two stages, we introduce a locate-informed retrospective reward, which uses LPR to guide and optimize the exploration behavior of GSE via reinforcement learning. The synergy between the coarse-to-fine framework and reinforcement learning not only enables fine-grained perception of extremely small objects but also allows for data-efficient multi-task training, resulting in a unified framework that consistently delivers high performance across both reasoning and segmentation tasks.

To enable a comprehensive evaluation, we introduce FINERS-4k, a human-annotated, high-quality dataset designed to benchmark model performance on Instruction-guided Segmentation (IS), Open-ended Visual Question Answering (OVQA), and Multiple-choice Visual Question Answering (MVQA). Compared to previous high-resolution benchmarks [9, 10], FINERS-4k leverages UAV-captured imagery, providing large-scale, complex environments with extreme object size variability, scattered small-object distributions, and cluttered spatial contexts. Extensive experiments on FIN-ERS-4k and other public datasets [9, 10] demonstrate that FINERS consistently outperforms existing MLLM-based methods in both answer accuracy and segmentation precision.

To summarize, our contributions are as follows:

- We propose FINERS, a two-stage MLLM framework that jointly performs instruction-guided reasoning and segmentation for small-object understanding in high-resolution images. To the best of our knowledge, FINERS is the first method to unify reasoning and fine-grained segmentation under a reinforcement learning paradigm.

- We introduce FINERS-4k, the first UAV-captured high-resolution dataset designed for instruction-guided reasoning and segmentation on ultra-small objects, offering more challenging object distributions and spatial variability compared to previous datasets.

- We conduct extensive experiments on FINERS-4k and other public datasets, demonstrating that the proposed FINERS consistently outperforms state-of-the-art MLLM-based approaches across instruction-guided segmentation, open-ended VQA, and multiple-choice VQA.

## 2 Related Works

**MLLM-based Reasoning and Segmentation.** The success of MLLMs has significantly advanced object detection and segmentation for more accurate open-world understanding. Pioneering efforts such as LISA [3] and LISA++ [17] introduce a `<SEG>` token to bridge MLLMs with segmentation models [7]. To evaluate performance, they propose reasoning segmentation, an extension of referring segmentation that enables simultaneous generation of text responses and segmentation masks. This design has inspired a series of follow-up works [18, 19], which further explore special-token-based interfaces to integrate vision-language reasoning with segmentation. Despite their promising results, these approaches still face notable limitations. First, most MLLM-based models restrict the input resolution to avoid out-of-memory risks, leading to severe downsampling that compromises fine-grained visual details and degrades performance on high-resolution imagery. Second, they rely heavily on supervised training with extensive public datasets [20, 3], which limits their adaptability and transferability to more challenging scenarios with limited training data availability. In this work, we aim to address the challenges of small-object reasoning and segmentation within high-resolution images. We propose a two-stage MLLM framework that combines global semantic exploration with localized perceptual refinement and applies a reinforcement learning strategy to achieve data-efficient fine-tuning.

**High-resolution Image Understanding and Reasoning.** Recent studies [9, 10, 1, 21] have revealed that MLLMs still face significant challenges in perceiving and reasoning over high-resolution images, particularly for small and densely distributed objects. To overcome the resolution restrictions of MLLMs, fine-tuning-based methods [22, 23] divide input images into uniform patches and process them in parallel with visual encoders, enabling MLLMs to handle arbitrary-resolution inputs. Concurrently, SEAL [9] considers a more complex challenge of small object perception in high-resolution images. It introduces both an evaluation benchmark and a guided visual search mechanism that leverages LLM priors to selectively focus on important regions, effectively improving visual reasoning ability in complex and crowded high-resolution scenarios. Due to the scarcity of available training data, subsequent methods [10, 21] have developed training-free pipelines that apply hierarchical image partitioning to form stepwise reasoning processes. Similarly, attention-based

Table 1: Comparisons of different benchmarks. The annotation type includes Question (Q), Answer (A), and object mask. The supported tasks are Multiple-choice Visual Question Answering (MVQA), Open-ended Visual Question Answering (OVQA), and Instruction-guided Segmentation (IS). For small object granularity, "Partial" means that the dataset contains some small objects but doesn't provide an explicit indication.

| Dataset | HR Images | Annotation Type | Sample Num | Supported Tasks | Small Object Granularity |
|---|---|---|---|---|---|
| V* [9] | ✓ | Q&A | 191 | MVQA | ✗ |
| HR-Bench [10] | ✓ | Q&A | 200 | MVQA | ✗ |
| refCOCOg [20] | ✗ | Q&Mask | 95,010 | IS | Partial |
| ReasonSeg [3] | ✗ | Q&Mask | 1,218 | IS | Partial |
| **Ours (FINERS-4k)** | ✓ | **Q&A&Mask** | **12,132** | **MVQA&OVQA&IS** | **S/XS/XXS** |

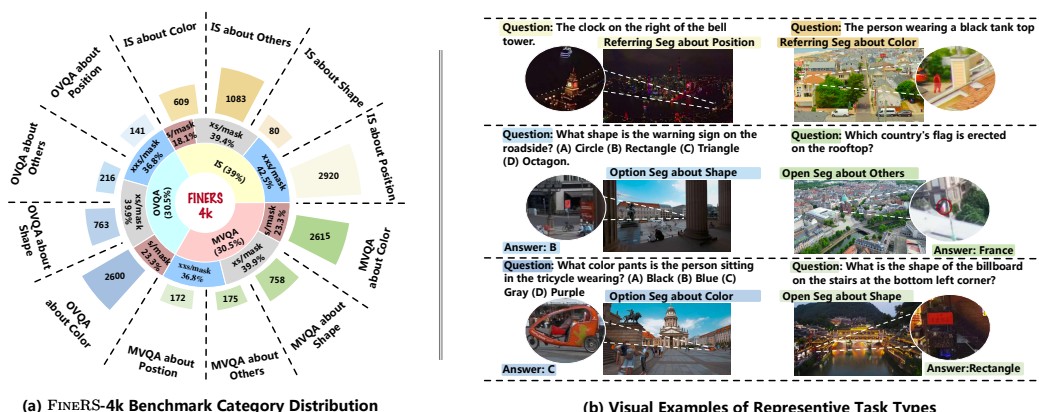

(a) FINERS-4k Benchmark Category Distribution

(b) Visual Examples of Representative Task Types

Figure 2: Overview of our benchmark. (a) The innermost ring shows three instruction types. The middle ring presents the mask size distribution within each type. The outermost ring breaks each type down into four attribute categories: color, shape, position, and others. (b) Visualization of six representative examples, each illustrating a different attribute category (color, shape, position, others) across the three instruction types in the outermost ring.

visual intervention methods [1] enhance the perception of small visual details by interpreting and manipulating internal attention maps of MLLMs. Despite these advancements, existing methods remain sensitive to heuristic cropping algorithms and still fail to achieve precise object localization. In this paper, we introduce a large-scale dataset and propose a data-efficient fine-tuning pipeline that employs reinforcement learning to adapt MLLMs for small object reasoning and segmentation.

**Reinforcement Learning for MLLMs.** Recently, Reinforcement Learning (RL) [24, 25] has become a new-emerging technique for enhancing reasoning in large language models, as demonstrated by OpenAI's o1 [11] and DeepSeek R1-Zero [12]. Among them, a critic-free algorithm, Group Relative Policy Optimization (GRPO) [13] is designed to eliminate Supervised Fine-Tuning (SFT) by directly comparing candidate responses in groups. Inspired by this, Visual-RFT [14] proposes an RL-based fine-tuning strategy for Large Vision-Language Models (LVLMs), improving performance on classification and detection tasks using GRPO-based rewards under limited supervision. However, it is limited to coarse-level tasks such as classification and detection, and does not support fine-grained segmentation. In contrast, Seg-Zero [16] leverages high-quality box-level rewards within an RL framework and feeds them into a frozen SAM2 [8] segmentation model, enabling pixel-level visual perception and reasoning. While it performs well on standard-resolution and regular object scenarios, challenges remain in small-object segmentation for high-resolution images. Moreover, Seg-Zero adopts a fixed reward paradigm, making it difficult to generalize to more diverse question-answering formats, such as open-ended and multiple-choice VQA. Inspired by the success of RFT in vision-language tasks, we apply RFT to our two-stage MLLM framework, where both localization and VQA rewards are integrated to boost the unification of object segmentation and VQA. Besides, we introduce a retrospective reward between two stages for more consistent global-to-local perception.

## 3 FINERS-4k Benchmark

To comprehensively evaluate the capacities of MLLMs for instructed-guided ultra-small object reasoning and segmentation, we construct a new dataset FINERS-4k. Unlike previous high-resolution benchmarks [9, 10] that are primarily captured by handheld or ground-based cameras under structured conditions, FINERS-4k comprises images captured by Unmanned Aerial Vehicles (UAVs). This enables a much wider field-of-view and introduces complex visual challenges such as dense layouts, extreme variations in object scale, and small-object sparsity.

We begin by collecting 4k-resolution drone videos ($3840 \times 2060$) from YouTube and our own UAV footage. Volunteers are then tasked with filtering high-quality frames and annotating small objects with a triplet annotation consisting of question, answer, and mask. Considering the difficulty in annotating ultra-small objects in high resolution images, annotators were primarily instructed to identify a single, unambiguous small object of interest from the image and compose questions that uniquely specify it (e.g., referencing color, shape, position, or context). For several cases with multiple similar objects, they were required to formulate disambiguating questions and to visually inspect the entire image to ensure no other object matched the same description. The annotation was completed by 14 volunteers, organized in pairs for mutual cross-checking. In addition, a team of 4 senior reviewers conducted a final round of quality assurance to correct ambiguities and verify consistency. This multi-stage validation process was designed to maximize precision and minimize annotation bias. This process results in 8,411 annotated small entities across 4,563 high-resolution images, yielding a total of 12,132 text-mask pairs. Specifically, we divide them into train set (8,956), validation set (749), and test set (2,427). The overall comparison of our FINERS-4k and other datasets is illustrated in Tab. 1.

Fig. 2 illustrates a detailed analysis of the distribution of task type and object sizes. As shown in the innermost ring of Fig. 2 (a), FINERS-4k provides instructions for three sub-tasks, including **1) Instruction-guided Segmentation (IS, 39%)** that requires generating a mask based on the instruction; **2) Multiple-choice VQA (MVQA, 30.5%)** that involves predicting both a segmentation mask and an option based on the option-given instruction; and **3) Open-ended VQA (OVQA, 30.5%)** that requires producing both a mask along and a free-form answer. Each entity is bound to at least one instruction type. We further classify object size into three categories based on their proportion to the entire image area: small (S, >0.055%), extra small (XS, 0.017%–0.055%), and extra-extra small (XXS, < 0.017%). The distribution of these object sizes across task types is shown in the second ring of Fig. 2 (a). The outermost ring illustrates the types of attribute-specific instructions, including color, shape, position, and other distributions. Combining the three task types and four attribute types yields 12 distinct instruction-task combinations, with the numbers indicating the annotated instance count for each sub-task. Fig. 2 (b) illustrates the visual examples of different tasks in FINERS-4k. More detailed analysis about object size and spatial distribution can be found in Fig. A1.

## 4 Methodology

### 4.1 Preliminary

**Task Definition.** Given an image $I$ and a user instruction $Q$, MLLMs [4, 2] are capable of jointly understanding visual and textual input to generate appropriate responses $A^{pre}$. The objective of instruction-guided segmentation [3, 16] is to predict a segmentation mask $M^{pre}$ based on the image $I$ and instruction $Q$. In contrast, our method unifies these two tasks into a single framework, $(I, Q) \to (A^{pre}, M^{pre})$, enabling simultaneous instruction-guided segmentation, open-ended VQA, and multiple-choice VQA.

**GRPO for Visual Perception.** Visual-RFT [14] and Seg-Zero [16] extend the GRPO [13] framework to visual perception tasks by introducing task-specific rewards. Given an input image $I$ and instruction $Q$, the model generates $n$ candidate predictions of the expected output, each of which is compared against ground-truth coordinates to compute individual rewards. GRPO then performs group-wise normalization over these rewards, guiding the model to favor perceptually accurate outputs, even in the absence of reasoning data during cold-start training. This approach enables more effective visual alignment and improves the generalization capability of MLLMs for object detection, classification, and referring segmentation.

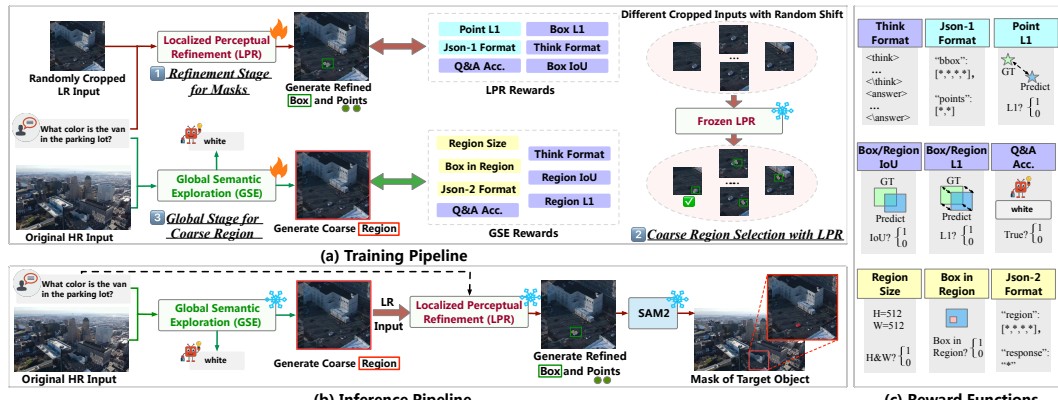

Figure 3: Framework overview of FINERS. (a) During training, we design specific reward functions to train GSE and LPR, where the LPR is optimized first and adopted to form a retrospective reward to enhance the coarse region accuracy of GSE. (b) During inference, GSE takes a high-resolution image and user instructions as input and produces an answer and a coarse region containing referred objects. Then, LPR processes the instruction and coarse region to generate the object box and adopts SAM2 [8] to generate the final mask. (c) To unify VQA and segmentation into a single MLLM, we design a multi-task reward pool and assign the items to supervise GSE and LPR.

## 4.2 Overall Framework of FINERS

The overall framework of FINERS is illustrated in Fig. 3. It consists of a two-stage MLLM pipeline, comprising Global Semantic Exploration (GSE) and Localized Perceptual Refinement (LPR). Unlike prior high-resolution MLLMs [10, 21] that rely on complex search strategies across multiple cropped image patches, our framework is designed to produce both textual responses and object segmentation masks through a single feedforward pass. As shown in Fig. 3 (a), during training, GSE and LPR are optimized independently with specially designed rewards to facilitate perception at different levels of granularity. We introduce a locate-informed retrospective reward, which leverages LPR to select a robust coarse region, enhancing the global exploration precision of GSE. The detailed reward formulations for both modules are shown in Fig. 3 (c). During inference, as shown in Fig. 3 (b), only the original high-resolution image and user instruction are fed into GSE, which directly outputs the coarse box $B_r^{pre}$ and the text response $A^{pre}$. The GSE stage can be formulated as:

$$(B_r^{pre}, A^{pre}) = \mathcal{G}(\theta_{\text{GSE}}(I, Q)), \tag{1}$$

where $\mathcal{G}$ represents a post-processing function to extract the keywords of the long response. To constrain the search space of $B_r^{pre}$, we set a fixed box size to optimize only the center offsets. The coarse box indicates an enlarged region around the referred object, which will subsequently be passed to LPR for object localization and mask generation. In the second stage, LPR first crops the original image based on $B_r^{pre}$ to obtain a lower-resolution input $I_c$, and performs local reasoning to generate bounding boxes $B^{pre}$ and points $P_1^{pre}, P_2^{pre}$. The LPR stage can be formulated as:

$$(B^{pre}, P_1^{pre}, P_2^{pre}) = \mathcal{G}(\theta_{\text{LPR}}(I_c, Q)). \tag{2}$$

Finally, the generated boxes and points are passed to the frozen SAM2 [8] to produce the segmentation mask $M^{pre}$ for the target object.

## 4.3 Training Pipeline

The comparison in Tab. 1 illustrates that the number of samples in existing high-resolution benchmarks is substantially lower than that of standard-resolution datasets. To avoid data overload of supervised fine-tuning, we exploit a recent vision reinforcement learning strategy to enhance the reasoning capacities of two-stage MLLMs. The training process is illustrated in Fig. 3 (a). Specifically, we train our FINERS model with the GRPO algorithm in two stages. First, the coarse-grained LPR module is trained with LPR rewards to generate the object box from a local image crop. Then, the GSE module is optimized using a combination of GSE rewards and a locate-informed retrospective reward guided by the LPR module. In addition, a standard KL divergence penalty is applied between the policy and the reference model.

❶ **Refinement Stage for Masks.** The supervised annotations for LPR training include bounding boxes $B^{gt}$, points $P_1^{gt}$ and $P_2^{gt}$, and text responses $A^{gt}$, all derived from the masks and answer annotations provided in training data. The <think> process originates from a prompt-based cold start and is supervised using the <think> format. Notably, since LPR is designed to focus on fine-grained perception within lower-resolution regions, it is trained on image patches that are randomly cropped around the ground-truth bounding boxes.

❷ **Coarse Region Selection with LPR.** Unlike LPR, the coarse region of GSE lacks explicit ground truth supervision for reward calculation. To address this, we design a locate-informed retrospective reward that uses the outputs of LPR to provide robust coarse region supervision for GSE. Specifically, for each training sample, we first generate $n$ randomly offset coarse regions that cover the GT bounding box $B^{gt}$. We then compute the IoU score between the LPR-predicted boxes $B^{pre}$ and the ground-truth boxes, selecting the region with the highest IoU as the GT regions $B_r^{gt}$ for training the GSE module.

❸ **Global Stage for Coarse Region.** Due to the complex scenes in high-resolution images, which make it challenging for the model to focus on small targets, the GSE model is designed to generate approximate regions where small targets are likely to exist, based on the instruction context. Therefore, the training data for GSE consists of high-resolution images annotated with the optimal regions $B_r^{gt}$ selected by LPR and corresponding ground-truth answer labels $A^{gt}$.

## 4.4 Reward Functions

Inspired by the reward functions of the GRPO strategy, we design distinct reward functions at different levels of granularity for the LPR and GSE modules.

**Rewards for LPR.** The rewards for LPR include Point L1 $R_{point}$, Box L1 $R_{bL1}$, Box IoU $R_{bIoU}$, JSON-1 format $R_{format1}$, Think format $R_{think}$, and Q&A Accuracy $R_{response}$. Among these functions, $R_{point}$, $R_{bL1}$, and $R_{bIoU}$ are computed based on the predicted boxes $B^{pre}$, predicted points $P^{pre}$, GT boxes $B^{gt}$ and GT points $P^{gt}$, to enforce spatial alignment between predictions and ground-truth annotations. The JSON-1 format reward $R_{format1}$ is only considered correct if the model outputs exact keywords {bbox, points 1, points 2, response} in the required structure. The response reward $R_{response}$ for the final response $A^{pre}$ is defined as:

$$R_{response} = \begin{cases} 1 & , \quad \text{if } A^{pre} \text{ is True,} \\ 0 & , \quad \text{if } A^{pre} \text{ is False,} \end{cases} \tag{3}$$

where the criteria for determining whether $A^{pre}$ is correct vary across task settings. For instruction-guided segmentation, the response is considered correct if it includes phrases like "is detected" or "is found". In the multiple-choice VQA setting, the response is correct if it exactly matches the ground-truth option. In the open-ended VQA setting, the response is deemed correct if the fuzzy matching similarity to the ground-truth answer exceeds 0.8. The final reward of LPR is computed as:

$$R_{LPR} = R_{bIoU} + R_{bL1} + R_{point} + R_{format1} + R_{response} + R_{think}. \tag{4}$$

**Rewards for GSE.** Unlike LPR, the reward functions for GSE are designed to encourage alignment between the predicted coarse region $B_r^{pre}$ and the ground-truth region $B_r^{gt}$ selected by LPR. Specifically, we name this reward as locate-informed retrospective reward that consists of a region IoU reward $R_{regionIoU}$ and a region L1 $R_{regionL1}$ between the predicted and GT regions $B_r^{pre}$, $B_r^{gt}$. Since this stage focuses solely on contextual regions rather than fine-grained localization, the point-level reward is omitted, and the output JSON format is updated to a new template as {region, response}. Additionally, to ensure that the predicted regions are compatible with the input size expected by the LPR module and sufficiently cover the target object, we introduce a region size $R_{size}$ reward and a box-in-region $R_{cover}$ reward. They encourage the model to generate regions of appropriate size and position, aligned with those used during LPR training. The think format and Q&A accuracy rewards are kept consistent with those used in LPR. The final reward of GSE is computed as:

$$R_{GSE} = R_{regionIoU} + R_{regionL1} + R_{size} + R_{cover} + R_{format2} + R_{response} + R_{think}. \tag{5}$$

Table 2: Performance comparison on the test set of FINERS-4k. "†" indicates that the corresponding method is retrained with our dataset. We label the best results with a **bold** style.

| Method | IoU (gIoU/cIoU) | | | | QA Acc. (MVQA/OVQA) | | | | |
|---|---|---|---|---|---|---|---|---|---|
| | *S* | *xS* | *xxS* | *All* | *Color* | *Shape* | *Others* | *Position* | *All* |
| *Training-free* | | | | | | | | | |
| LISA 7B [3] | 19.1/6.49 | 8.28/1.20 | 4.19/0.34 | 9.00/2.38 | 0.00/6.11 | 0.00/0.00 | 0.00/9.37 | 0.00/16.7 | 0.00/5.51 |
| LISA 13B [3] | 16.4/3.86 | 7.02/0.73 | 2.55/0.18 | 7.29/1.42 | 0.00/6.46 | 0.00/6.55 | 0.00/9.37 | 0.00/5.55 | 0.00/6.58 |
| LISA++ 7B [17] | 25.9/12.3 | 13.5/2.90 | 3.70/0.70 | 12.3/5.20 | 5.90/9.79 | 0.82/1.63 | 35.7/6.24 | 26.3/5.55 | 6.72/8.19 |
| PixelLM 7B [19] | 13.6/6.70 | 3.30/1.10 | 0.50/0.10 | 4.40/2.10 | 0.00/4.56 | 0.82/0.09 | 3.57/9.31 | 0.09/5.55 | 0.27/4.05 |
| SEAL [9] | – | – | – | – | 7.46/3.14 | 8.26/0.03 | 7.14/15.6 | 5.26/16.7 | 7.53/3.49 |
| DC$^2$ [10] | – | – | – | – | 39.0/19.5 | 36.3/9.00 | 57.1/21.8 | 36.8/16.6 | 39.2/17.8 |
| MLLMs-Know 7B [1] | – | – | – | – | 52.8/50.2 | 45.5/32.0 | 60.7/34.4 | 36.8/5.56 | 51.5/45.4 |
| MLLMs-Know 13B [1] | – | – | – | – | 56.1/54.0 | 36.4/32.0 | 67.9/37.5 | 26.3/16.7 | 52.6/48.8 |
| MLLMs-Know 7B [1] + LISA 7B [3] | 21.2/19.2 | 15.0/7.95 | 11.3/4.50 | 14.9/12.5 | 52.8/50.2 | 45.5/32.0 | 60.7/34.4 | 36.8/5.56 | 51.5/45.4 |
| MLLMs-Know 13B [1] + LISA 13B [3] | 27.0/20.3 | 18.3/9.29 | 12.4/3.62 | 17.9/12.8 | 56.1/54.0 | 36.4/32.0 | 67.9/37.5 | 26.3/16.7 | 52.6/48.8 |
| Seg-Zero 7B [16] | 55.9/18.6 | 34.7/4.94 | 16.5/0.84 | 32.1/6.61 | – | – | – | – | – |
| *Training* | | | | | | | | | |
| LISA$^†$ 7B [3] | 13.0/9.62 | 15.0/11.4 | 8.78/5.59 | 12.1/9.64 | 23.9/24.3 | 24.8/18.6 | 25.0/6.2 | 21.1/0.0 | 23.9/22.0 |
| PixelLM$^†$ 7B [19] | 1.27/1.02 | 0.52/0.35 | 0.08/0.02 | 0.16/0.13 | 0.0/0.0 | 0.0/0.0 | 0.0/0.0 | 0.0/0.0 | 0.0/0.0 |
| MLLMs-Know 7B [1] + LISA$^†$ 7B [3] | 1.32/1.16 | 2.08/1.90 | 2.85/3.15 | 2.22/1.57 | 52.8/50.2 | 45.5/32.0 | 60.7/34.4 | 36.8/5.56 | 51.5/45.4 |
| Seg-zero$^†$ 7B [16] | 61.8/50.5 | 53.0/30.2 | 31.7/20.7 | 46.6/38.6 | – | – | – | – | – |
| **Ours (FINERS) 7B** | **62.2/52.6** | **59.0/43.1** | **47.2/27.5** | **55.1/46.5** | **85.8/60.5** | **76.0/49.2** | **78.6/34.4** | **63.2/27.8** | **83.3/56.7** |

## 5 Experiments

### 5.1 Experimental Settings

**Implementation Details.** Our two-stage MLLMs are built upon Qwen2.5-VL-7B [2], with input resolution of $1920 \times 1080$ for GSE and $512 \times 512$ for LPR. The output coarse region of GSE is $256 \times 256$, which will be $2\times$ upsampled before being fed into LPR. The whole model is trained on a $4\times$A800 GPU (80G) setup using the Seg-Zero [16] and DeepSpeed [12] library. During training, the GSE module uses a total batch size of 16 with 8 samples per training step, while the LPR module uses a total batch size of 32, also with 8 samples per step. For both stages, the initial learning rate is set to 1e-6 and the weight decay is 0.01. In addition, we adopt SAM2 [8] for box-to-mask generation, which is kept frozen during training. The user prompts for GSE and LPR across three tasks are presented in Fig. A3 and Fig. A4.

**Evaluation Metrics.** Following previous works [26, 27], we calculate gIoU and cIoU for instruction-guided segmentation. The gIoU is the average of all per-image Intersection-over-Unions (IoUs), while the cIoU calculates the cumulative intersection over the cumulative union. We evaluate cIoU and gIoU metrics across different object sizes. In addition, we calculate the accuracy of multiple-choice (MVQA) and open-ended (OVQA) visual question answering using option accuracy and the "difflib.SequenceMatcher" algorithm, with a matching threshold set to 0.8.

### 5.2 Comparison with State-of-the-art Methods

We evaluate FINERS and other methods on our FINERS dataset across three tasks, including instruction-guided segmentation, open-ended VQA, and multiple-choice VQA.

❶ **Comparison on FINERS-4k.** The comparison results on the test and validation sets of our FINERS-4k are illustrated in Tab. 2 and Tab. A1, respectively. We report both training-free approaches and selected retrained methods in our dataset. As shown, our model consistently outperforms state-of-the-art segmentation approaches and high-resolution VQA methods. Note that the IoU scores are computed on all samples from the three instruction types, while QA accuracy is calculated only on the samples of MVQA and OVQA. The performance drop of LISA [3] and PixelLM [19] likely arises from the domain shift and task complexity of FINERS-4k, which contains 4K UAV imagery with ultra-small, sparse objects. All baselines are fine-tuned on FINERS-4k under the same settings. In this low-data, high-resolution regime, fixed-architecture models tend to overfit or underfit.

❷ **Comparison on Other VQA Datasets.** We also conduct a comparison with other approaches on public high-resolution VQA datasets, including V* [9] and HR-Bench [10]. Compared with our benchmark, these datasets are captured in a general, non-UVA perception with optional VAQ annotations. As shown in Tab. 3, without an additional finetuning process, our model achieves

Table 3: Performance comparison on other high-resolution VQA benchmarks. "*Attr.*" and "*Spat.*" denote attribute and spatial, respectively. We label the best methods with a **bold** style.

| Method | Segmentation | V* | | | HR-Bench 4K | | | HR-Bench 8K | | |
|---|---|---|---|---|---|---|---|---|---|---|
| | | *Attr.* | *Spat.* | *Overall* | *FSP* | *FCP* | *Avg.* | *FSP* | *FCP* | *Avg.* |
| SEAL [9] | ✗ | 74.8 | 76.3 | 75.4 | 47.0 | 29.3 | 38.1 | 42.5 | 28.8 | 35.6 |
| Mllms-Know [1] | ✗ | – | – | 62.3 | 52.4 | 30.2 | 41.3 | 47.2 | 30.7 | 38.9 |
| VILA-HD-1.5K [28] | ✗ | – | – | 68.1 | – | – | – | – | – | – |
| VILA-HD-4K [28] | ✗ | – | – | 71.2 | – | – | – | – | – | – |
| DC$^2$ [10] | ✗ | – | – | 57.3 | 53.0 | 47.0 | 50.0 | 37.2 | 44.2 | 40.8 |
| **Ours (FINERS)** | ✓ | **76.5** | **79.0** | **77.5** | **66.4** | **61.2** | **63.8** | **60.2** | **55.9** | **58.1** |

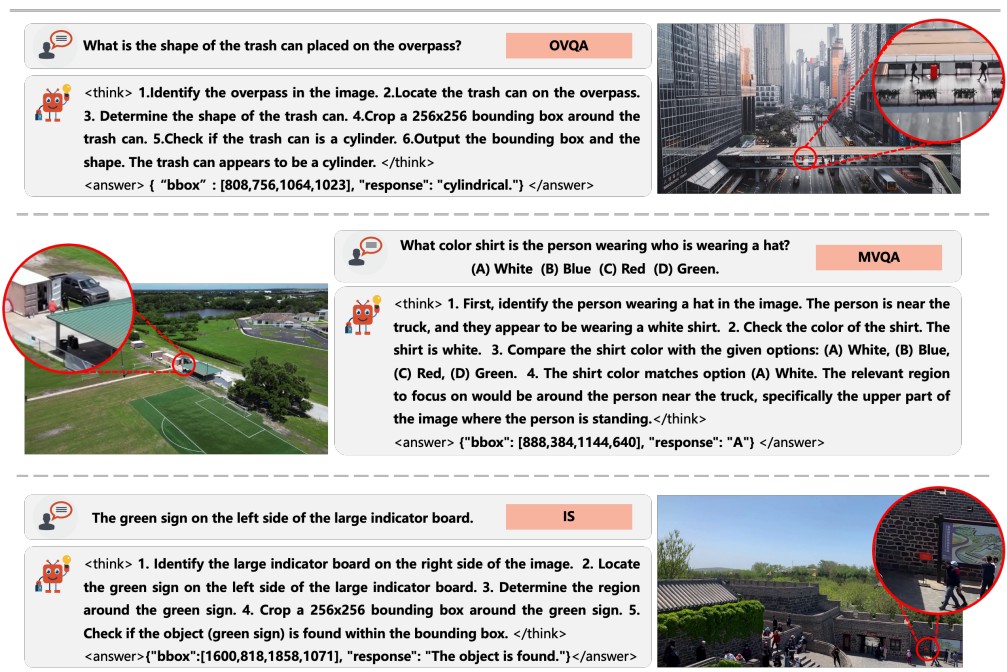

Figure 4: Visual results on Open-ended VQA (OVQA), Multiple-choice VQA (MVQA), and Instruction-guided Segmentation (IS). The listed images are sampled from FINERS-4k test set.

significantly better accuracy in non-UVA scenarios from 4k to 8k resolutions. Note that SEAL [9] is the model proposed in the V* benchmark.

❸ **Visualization Results.** Representative visual results for all three tasks are provided in Fig. 4. As shown, the proposed two-stage framework is effective in locating tiny objects from a cluttered background and generating accurate open-ended or optional answers from the textual instructions.

## 5.3 Ablation Study

Tab. 4 illustrates the evaluation results by separately removing the key components from the whole framework. To validate the efficacy of retrospective reward, we replace the LPR-informed coarse regions with a random crop centered around the GT object to supervise the GSE for coarse box generation. The result of "*w/o Restrospective Reward*" demonstrates the effectiveness of this design. To mitigate the sensitivity of LPR to input box variations, we apply a box augmentation strategy during LPR training. The improvement observed in "*w/o Random Input Region in LPR*" verifies the efficacy of this data augmentation strategy. In contrast to [16], our FINERS can generate exact text answers to user questions, which requires the incorporation of a QA accuracy reward during training. The performance drop in "*w/o QA Acc. Reward*" supports the importance of this component. More detailed analysis on this reward can be found in Tab. A2. In GSE, we use extra box rewards to facilitate the optimization of coarse region generation. The ablation results of "*w/o Box Size Reward*" and "*w/o Box-in-region Reward*" demonstrate the positive impact of these rewards on model performance.

Table 4: Ablation studies about the proposed components on FINERS-4k.

| Different Settings | Test set | | | | Val set | | | |
|---|---|---|---|---|---|---|---|---|
| | *gIoU* | *cIoU* | *MVQA* | *OVQA* | *gIoU* | *cIoU* | *MVQA* | *OVQA* |
| **FINERS** | **55.1** | **46.5** | **83.3** | **56.7** | **49.9** | **39.4** | **87.2** | **60.0** |
| *w/o Retrospective Reward* | 54.0 | 44.0 | 82.3 | 53.0 | 49.4 | 38.0 | 86.2 | 55.8 |
| *w/o Random Input Region in LPR* | 53.9 | 46.7 | 83.7 | 61.9 | 48.7 | 39.4 | 84.8 | 59.4 |
| *w/o QA Acc. Reward* | 52.8 | 45.7 | – | – | 48.6 | 33.7 | – | – |
| *w/o Box Size Reward* | 51.0 | 43.4 | 56.5 | 40.8 | 44.5 | 35.8 | 78.4 | 56.4 |
| *w/o Box-in-region Reward* | 50.1 | 42.0 | 56.5 | 40.8 | 44.1 | 34.2 | 77.9 | 55.7 |

## 6 Discussion

In this paper, we aim to resolve the fine-grained reasoning and segmentation of ultra-small objects in high-resolution images. We introduce 1) FINERS, a two-stage MLLM-based reinforcement learning framework that combines global semantic exploration with localized perceptual refinement; and 2) FINERS-4k, a new dataset featuring challenging scenes annotated with text-mask pairs across three types of tasks. Extensive experiments on FINERS-4k and other public benchmarks demonstrate the superiority of the proposed method in both answering accuracy and segmentation precision.

Although our method made an early attempt to apply reinforcement learning for joint object reasoning and segmentation, several limitations remain in the current architecture. First, the localization accuracy of LPR is highly dependent on the output coarse region of GSE. The text answer and object box will be incorrect if the GT object exceeds the coarse region. Incorporating an additional signal for object existence and enabling re-exploration could help mitigate such missed detection errors. Second, due to memory constraints, the two-stage models are not jointly optimized during training. Designing a more efficient architecture for jointly end-to-end training of two stages remains an important future direction.

## Acknowledgment

This work was supported by 2024-0011 (ZX20240867).

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

# A Appendix

In this section, we provide additional figures and tables of the analysis on the proposed method and benchmark.

## A.1 More Analysis on FINERS-4k

In this paper, we introduce a new dataset, FINERS-4k, which consists of high-resolution images containing ultra-small objects with diverse spatial distributions. Fig. A1 illustrates the detailed distribution of object sizes and spatial locations across all samples in the training, validation, and test sets. As shown, our dataset exhibits more challenging scenarios with extra-small objects and sparse distributions.

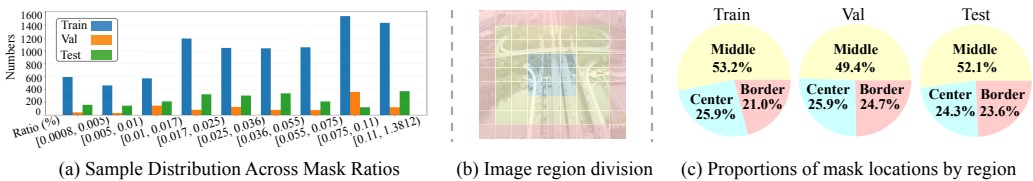

(a) Sample Distribution Across Mask Ratios    (b) Image region division    (c) Proportions of mask locations by region

Figure A1: (a) Distribution of mask sizes across three subsets, where the x-axis indicates the ratio of the mask area to the entire image. (b) Spatial division of the image into three regions — center, middle, and border—based on the location of the mask. (c) Distribution of mask locations across three subsets.

## A.2 Rewards Definition Details

In Eq. 4 and Eq. 5, we introduce several rewards for LPR and GSE modules. Specifically, the detailed definitions of each rewards in Eq. 4 are:

- $R_{point} = 1$ if L1 distance between predicted point and GT point is less than 100 pixels.
- $R_{bL1} = 1$ if the L1 distance between the predicted box and GT box is less than 10 pixels.
- $R_{bIoU} = 1$ if their IoU is greater than 0.5.
- $R_{response}$ is set to 1 if the predicted answer is correct (exact match for multiple choice, fuzzy match for open-ended QA).
- $R_{format}$ and $R_{think}$ are binary rewards that verify whether the output adheres to the expected JSON and reasoning formats.

In Eq. 5:

- $R_{region^{L1}} = 1$ if the L1 distance between the predicted coarse box $B_r^{pre}$ and GT region $B_r^{gt}$ is less than 10 pixels.
- $R_{region^{IoU}} = 1$ if their IoU is greater than 0.5; $R_{size}$ is 1 when the predicted coarse box is of size $512 \times 512$.
- $R_{cover}$ is 1 when the ground-truth object lies fully inside the predicted region.

We assign equal weights to all binary terms, following standard GRPO practices [16, 14], which yielded stable performance without tuning.

## A.3 More Comparison Results

Tab. A1 illustrates the comparison results on the validation set of FINERS-4k. While our method performs slightly lower than SegZero on small-sized objects, it significantly outperforms SegZero on xs and xxs objects. More importantly, our method supports VQA tasks and outperforms other approaches in this setting. Additional qualitative results produced by our method are shown in Fig. A2.

Table A1: Performance comparison on the validation set of FINERS-4k. "†" indicates that the corresponding method is retrained with our dataset. We label the best results with a **bold** style.

| Method | IoU (gIoU/cIoU) | | | | QA Acc. (Option/Open) | | | | |
|---|---|---|---|---|---|---|---|---|---|
| | S | xS | xxS | All | Color | Shape | Others | Position | All |
| *Training-free* | | | | | | | | | |
| LISA 7B [3] | 14.3/2.91 | 6.40/1.06 | 3.54/0.36 | 6.58/1.65 | 0.00/15.1 | 0.00/0.00 | 0.00/0.00 | 0.00/16.7 | 0.00/11.6 |
| LISA 13B [3] | 12.1/2.42 | 4.10/0.58 | 1.21/0.14 | 4.28/1.10 | 0.00/19.4 | 0.00/2.22 | 0.00/0.00 | 0.00/16.7 | 0.00/15.2 |
| LISA++ 7B [17] | 25.0/7.10 | 8.80/2.10 | 2.30/0.60 | 8.91/3.72 | 4.86/16.7 | 4.86/0.00 | 4.00/7.80 | 8.33/16.7 | 5.99/13.2 |
| PixelLM 7B [19] | 10.5/3.31 | 3.30/1.00 | 0.42/0.13 | 3.31/1.60 | 0.00/3.78 | 0.00/2.22 | 0.00/0.00 | 0.00/0.00 | 0.00/3.21 |
| SEAL [9] | – | – | – | – | 2.16/9.68 | 6.98/0.00 | 0.00/0.00 | 8.33/33.3 | 3.20/7.80 |
| DC² [10] | – | – | – | – | 34.6/21.0 | 25.6/6.67 | 40.0/0.00 | 33.3/16.7 | 33.2/17.2 |
| MLLMs-Know 7B [1] | – | – | – | – | 46.5/50.0 | 44.2/37.8 | 80.0/38.5 | 41.7/50.0 | 47.2/47.2 |
| MLLMs-Know 13B [1] | – | – | – | – | 50.3/51.1 | 32.6/26.7 | 80.0/23.1 | 91.766.7 | 50.4/45.6 |
| MLLMs-Know 7B [1] + LISA 7B [3] | 17.1/11.2 | 8.94/4.64 | 8.98/2.78 | 10.4/7.07 | 46.5/50.0 | 44.2/37.8 | 80.0/38.5 | 41.7/50.0 | 47.2/47.2 |
| MLLMs-Know 13B [1] +LISA 13B [3] | 23.2/16.0 | 13.5/6.01 | 9.57/2.55 | 13.6/9.16 | 50.3/51.1 | 32.6/26.7 | 80.0/23.1 | 91.7/66.7 | 50.4/45.6 |
| Seg-zero 7B [16] | 56.5/24.6 | 28.0/3.75 | 13.8/1.41 | 27.0/7.49 | – | – | – | – | – |
| *Training* | | | | | | | | | |
| LISA† 7B [3] | 14.0/10.8 | 9.92/7.70 | 7.27/4.52 | 9.50/8.62 | 4.86/16.7 | 2.32/0.00 | 39.99/7.69 | 8.33/16.7 | 5.99/13.2 |
| PixelLM† 7B [19] | 1.27/1.02 | 0.52/0.35 | 0.08/0.02 | 0.16/0.13 | 0.0/0.0 | 0.0/0.0 | 0.0/0.0 | 0.0/0.0 | 0.0/0.0 |
| MLLMs-Know 7B [1] + LISA† 7B [3] | 1.10/0.72 | 1.44/1.40 | 1.77/1.84 | 1.52/1.05 | 46.5/50.0 | 44.2/37.8 | 80.0/38.5 | 41.7/50.0 | 47.2/47.2 |
| Seg-zero† 7B [16] | **67.2/63.8** | 45.8/15.8 | 30.2/16.9 | 42.9/31.3 | – | – | – | – | – |
| **Ours (FINERS) 7B** | 64.8/55.3 | **50.8/27.1** | **42.5/21.0** | **49.7/38.6** | **85.4/65.6** | **88.4/46.7** | **100/30.8** | **91.7/66.7** | **86.8/60.4** |

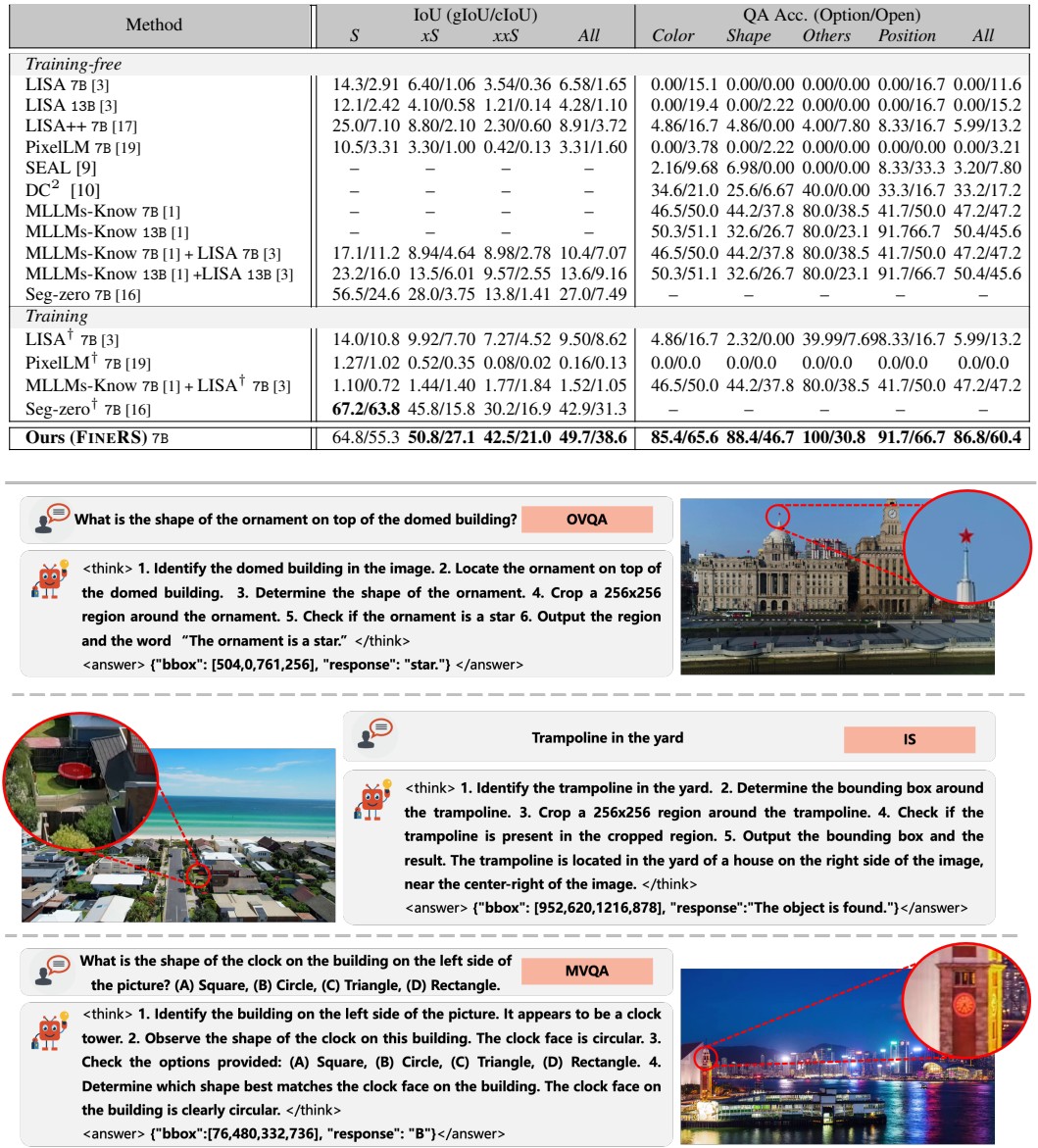

Figure A2: More visual results on Open-ended VQA (OVQA), Multiple-choice VQA (MVQA), and Instruction-guided Segmentation (IS). The listed images are sampled from FINERS-4k test set.

## A.4 More Ablation Studies

**QA Rewards in Different Stages.** Tab. A2 illustrates the ablation results on the effectiveness of QA accuracy reward and the two-stage designs. To evaluate the efficacy of our two-stage design, we remove the LPR stage and modify the GSE to directly generate object bounding boxes (i.e., $B^{pre}$ in Eq. 2. This baseline, "**One Stage** *xx*", can be seen as a higher-resolution version of SegZero [16]. From the comparison results, we can observe that the proposed two-stage coarse-to-fine framework is significantly effective in improving the localization precision of small objects. We also verify the effect of QA accuracy reward in GSE and LPR. The comparison results between "**GSE** *w/o QA Acc.* & **LPR** *w/o QA Acc.*" and "FINERS" demonstrate that the incorporation of additional VQA ability can effectively enhance the segmentation accuracy on small objects. Besides, the comparison between "**GSE** *w/ QA Acc.* & **LPR** *w/o QA Acc.*" and "**GSE** *w/o QA Acc.* & **LPR** *w/ QA Acc.*"

demonstrates that the textual response on GSE shows better performance due to the exploration of global visual information.

Table A2: Ablation study about QA accuracy reward.

| Method | Test set | | | | Val set | | | |
|---|---|---|---|---|---|---|---|---|
| | gIoU | cIoU | Option | Open | gIoU | cIoU | Option | Open |
| **One Stage** *w/o QA Acc.* | 46.6 | 38.6 | – | – | 41.6 | 31.3 | – | – |
| **One Stage** *with QA Acc.* | 42.3 | 41.8 | 84.4 | 58.2 | 40.1 | 37.6 | 88.5 | 63.6 |
| **GSE** *w/o QA Acc.* & **LPR** *w/o QA Acc.* | 52.8 | 45.7 | – | – | 48.6 | 37.7 | – | – |
| **GSE** *w/ QA Acc.* & **LPR** *w/o QA Acc.* | 53.7 | 46.2 | 83.3 | 56.7 | 49.2 | 38.2 | 87.2 | 60.0 |
| **GSE** *w/o QA Acc.* & **LPR** *w/ QA Acc.* | 54.4 | 46.0 | 56.2 | 39.1 | 48.1 | 38.2 | 55.6 | 40.4 |
| **FINERS** | **55.1** | **46.5** | 83.3 | 56.7 | **49.9** | **39.4** | 87.2 | 60.0 |

**Effects on Hyper-parameters.** In our experiments, all hyper-parameters are set to the default values in [16] without specific tuning. We conduct a sensitivity analysis on key hyper-parameters, including grouping $n$, KL weight, and seeds. Tab. A3 illustrates the representative hyper-parameter results on FINERS-4k test set, which are consistent with SegZero. For a fair comparison, we did not perform expensive hyper-parameter tuning.

Table A3: Hyper-parameter sensitivity analysis on FIN-ERS-4k test set.

| Hyper-parameters | gIoU | cIoU | Option | Open |
|---|---|---|---|---|
| **Group 8** | **55.1** | **46.5** | **83.3** | **56.7** |
| Group 6 | 54.3 | 43.7 | 82.6 | 53.5 |
| Group 4 | 52.4 | 43.9 | 82.9 | 55.7 |
| **KL 5e-3** | **55.1** | **46.5** | **83.3** | **56.7** |
| KL 5e-2 | 54.4 | 46.1 | 82.9 | 55.8 |
| **Seed 42** | **55.1** | **46.5** | **83.3** | **56.7** |
| Seed 48 | 53.9 | 45.4 | 82.0 | 56.2 |
| Seed 80 | 55.1 | 47.6 | 82.9 | 55.7 |

**Efficiency Comparison.** We report the average wall-clock inference time for 4k-resolution inputs in Tab. A4. All models were evaluated on a single A100 GPU with consistent runtime environments. As shown, compared to SEAL [9] and DC2 [10], our method and Seg-Zero [16] exhibit higher inference latency due to the use of CoT reasoning. Despite our two-stage framework requires extra inference time, this design is essential for achieving precise reasoning and segmentation of ultra-small objects in high-resolution scenes.

Table A4: Inference latency and performance of different methods.

| Method | FINERS-4k test set | | HR-bench 4k | |
|---|---|---|---|---|
| | gIoU/cIoU/MVQA/OVQA | Time (s/img) | QA Acc. | Time (s/img) |
| SEAL 7B [9] | –/–/7.53/3.49 | 1.21 | 38.1 | 1.15 |
| DC$^2$ 7B [10] | –/–/39.2/17.8 | 2.69 | 50.0 | 2.90 |
| SegZero$^\dagger$ 7B [16] | 46.4/38.6/–/– | 8.67 | – | – |
| Ours (FINERS) 7B | 55.1/46.5/83.3/56.7 | 7.31 (GSE) + 5.57 (LPR&SAM2) | 63.8 | 6.35 (GSE only) |

**Domain Generalization on ReasonSeg [3].** We evaluate our model on ReasonSeg in a zero-shot manner to verify its generalization on non-UVA scenarios. ReasonSeg contains ground-level daily scenes with moderate-to-large object sizes, significantly different from aerial, ultra-high-resolution imagery and ultra-small objects focus of FINERS-4k. We categorize test samples by mask-to-image area ratio into Large ( 50%), Middle (10–50%), and Small ( 10%). Notably, ReasonSeg's "Small" objects are still much larger than FineRS-4k's (<1%), introducing a challenging domain and scale gap. Despite this, as shown in Tab. A5, our two-stage model achieves better gIoU on Small objects, outperforming finetuned baselines like LISA$^\dagger$ and Seg-Zero$^\dagger$. This confirms the effectiveness of our coarse-to-fine strategy in segmenting small targets, even under mismatched resolution and context. Moreover, we find that our LPR shows strong adaptability across all sizes and outperforms other methods even without domain-specific finetuning, highlighting its robustness.

**Results on Other Baselines.** We evaluate our two-stage framework based on Qwen-3b [2], and the results are reported in Tab. A6. As observed, employing a smaller backbone leads to noticeable performance degradation, particularly on small objects. Nevertheless, our method still surpasses SegZero [16] under the same backbone configuration, demonstrating its superior adaptability and robustness.

Table A5: Performance on the ReasonSeg test set (IoU: gIoU / cIoU). Results are grouped by object size. Notely, "†" denotes the corresponding methods are finetuned with FineRS-4k without pretraining on large-scale referring segmentation datasets. "Resize" means we directly resize ReasonSeg image to meet our model's require ($1920 \times 1080$). "Padding" means that the low-resolution image are padded to meet our model's resolution.

| Method | Large (316 samples) | Middle (388 samples) | Small (72 samples) | ALL |
|---|---|---|---|---|
| Ours (FINERS) 7B (Resize) | 25.2 / 4.92 | 41.4 / 15.6 | 42.3 / 7.12 | 35.0 / 7.16 |
| Ours (FINERS) 7B (Padding) | 27.5 / 5.02 | 43.2 / 16.1 | 40.0 / 9.42 | 36.1 / 7.52 |
| LPR only | 59.0 / 52.4 | **54.4** / **35.3** | **45.7** / 24.6 | 56.6 / 42.1 |
| SegZero† 7B [16] | 49.3 / 41.7 | 46.1 / 30.9 | 35.8 / 15.4 | 47.1 / 38.8 |
| SegZero 7B [16] | **65.3** / **55.2** | 53.5 / 31.4 | 39.0 / 16.0 | **57.5** / **52.0** |
| LISA† 7B [3] | 0.44 / 0.34 | 4.08 / 1.38 | 8.14 / 1.12 | 2.98 / 0.55 |
| LISA 7B [3] | 55.3 / 56.7 | 34.2 / 26.9 | 20.3 / **24.8** | 48.7 / 48.8 |

Table A6: Performance comparison on the test set of FINERS-4k using Qwen2.5-VL (3b). "†" indicates that the corresponding method is retrained with our dataset.

| Method | IoU (gIoU/cIoU) | | | | QA Acc. (Option/Open) | | | | |
|---|---|---|---|---|---|---|---|---|---|
| | S | xS | xxS | All | Color | Shape | Others | Position | All |
| Seg-zero† 3B [16] | **57.7**/45.3 | 47.4/22.8 | 26.3/0.86 | 41.6/29.5 | – | – | – | – | – |
| **Ours (FINERS)** 3B | 57.3/**45.4** | **53.8/23.3** | **43.0/9.86** | **50.4/29.9** | 65.6/50.5 | 68.5/41.0 | 60.7/21.8 | 52.6/22.2 | 65.6/47.0 |

## A.5 User Prompt for FINERS

Fig. A3 and Fig. A4 illustrate the user prompts of Global Semantic Exploration (GSE) and Localized Proceptual Refinement (LPR) modules across three tasks, including Instruction-guided Segmentation (IS), Open-ended VQA (OVQA), and Multiple-choice VQA (MVQA).

---

**Prompt for IS**

" Based on the '{Question}', identify a 256*256 bounding box that best localizes the region most relevant to the query. And respond with whether the object is found. "
" Compare the difference between regions and find the most closely matched one. "
" Output the thinking process in <\think> and final answer in <\answer> <\answer> tags. "
" Output the 256*256 region bbox and the final response inside the interested object in JSON format. "
" i.e., <think> thinking process here <\think> "
" <answer>{'bbox' : [x_min,y_min,x_min+256,y_min+256] , response: 'The object is here.' }<\answer>"

---

**Prompt for OVQA**

" Based on the '{Question}', identify a 256*256 bounding box that best localizes the region most relevant to the query. And give me a final response with a word or phrase. "
" Compare the difference between regions and find the most closely matched one. "
" Output the thinking process in <\think> and final answer in <\answer> <\answer> tags. "
" Output the 256*256 region bbox and the final response inside the interested object in JSON format. "
" i.e., <think> thinking process here <\think> "
" <answer>'bbox' : [x_min,y_min,x_min+256,y_min+256], response: 'The cat is white.' }<\answer>"

---

**Prompt for MVQA**

" Based on the '{Question}', identify a 256*256 bounding box that best localizes the region most relevant to the query. And give me a correct option from {Options}. "
" Compare the difference between regions and find the most closely matched one. "
" Output the thinking process in <\think> and final answer in <\answer> <\answer> tags. "
" Output the 256*256 region bbox and the final response inside the interested object in JSON format. "
" i.e., <think> thinking process here <\think> "
" <answer>'bbox' : [x_min,y_min,x_min+256,y_min+256], response: 'A' }<\answer>"

Figure A3: User prompt for GSE module.

**Prompt for IS**

" Please find '{Question}' with bbox and points. **And respond with whether the target is found.**"

" Compare the difference between objects and find the most closely matched one. "

" Output the thinking process in <\think> and final answer in <\answer> <\answer> tags. "

" Output the one bbox and center points of two largest inscribed circles inside the interested object in JSON format. "

" i.e., <think> thinking process here <\think> "

" <answer>{'bbox' : [10,100,200,210], 'points 1' : [30,110], 'points 2' : [35,180] , **response: 'The object is here.'** }<\answer>"

**Prompt for OVQA**

" Please find '{Question}' with bbox and points. **And give me a final response with a word or phrase.**"

" Compare the difference between objects and find the most closely matched one. "

" Output the thinking process in <\think> and final answer in <\answer> <\answer> tags. "

" Output the one bbox and center points of two largest inscribed circles inside the interested object in JSON format. "

" i.e., <think> thinking process here <\think> "

" <answer>'bbox' : [10,100,200,210], 'points 1' : [30,110], 'points 2' : [35,180] , **response: 'The cat is white.'** }<\answer>"

**Prompt for MVQA**

" Please find ' {Question} ' with bbox and points. **And give me a correct option from {options}.**"

" Compare the difference between objects and find the most closely matched one. "

" Output the thinking process in <\think> and final answer in <\answer> <\answer> tags. "

" Output the one bbox and center points of two largest inscribed circles inside the interested object in JSON format. "

" i.e., <think> thinking process here <\think> "

" <answer>'bbox' : [10,100,200,210], 'points 1' : [30,110], 'points 2' : [35,180] , **response: 'A'** }<\answer>"

Figure A4: User prompt for LPR module.

