# OpenReview forum: "FineRS: Fine-grained Reasoning and Segmentation of Small Objects with Reinforcement Learning"
_NeurIPS.cc/2025/Conference — NeurIPS 2025 poster_

### Official Review · Reviewer_tyVp · 2025-07-02

**Clarity:** 3
**Significance:** 2
**Originality:** 2
**Rating:** 4
**Confidence:** 3

**Summary:**

The paper addresses instruction-guided reasoning and segmentation for high-resolution images by introducing the FINERS framework. This framework integrates the reinforcement learning (RL)-based approach Seg-Zero [1] with two modules for region search: Global Semantic Exploration (GSE) and Localized Perceptual Refinement (LPR). A new UAV-based benchmark dataset, FINERS-4k, is curated for evaluation. The proposed framework demonstrates superior performance on the FINERS-4k dataset as well as two other high-resolution VQA benchmarks, V* and HR-Bench, outperforming existing methods.

[1] Liu, Yuqi, et al. "Seg-zero: Reasoning-chain guided segmentation via cognitive reinforcement." arXiv preprint arXiv:2503.06520 (2025).

**Questions:**

1. Please explicitly highlight the key technical contributions of the proposed method in comparison to the Seg-Zero framework. If possible, demonstrating that the proposed modules (GSE and LPR) are also effective when applied to other frameworks or tasks would significantly strengthen the paper.

2. Please refer to Weakness 2 and address the concerns regarding the construction of the FINERS-4k dataset. Clarifying this aspect is crucial to ensure that no potential bias was introduced during the dataset collection process.

**Ethical Concerns:**

["NO or VERY MINOR ethics concerns only"]

**Final Justification:**

While I still view FINERS as an engineering solution that heavily builds upon Seg-Zero, the proposed extensions (e.g., GSE and LPR) effectively broaden its applicability. Although some concerns about potential bias in the FINERS-4k dataset remain, the dataset is still valuable if its limitations are properly discussed. I have decided to maintain my positive assessment of the paper.

**Limitations:**

Yes.

**Quality:**

3

**Strengths And Weaknesses:**

Strengths:

1. The curated FINERS-4k benchmark has the potential to be useful and valuable for various machine learning communities. It provides approximately 12k high-resolution (4K) images containing small objects, along with corresponding questions, answers, and mask annotations, supporting a wide range of tasks.

2. FINERS demonstrates promising performance across all tasks (instruction-guided segmentation, multiple-choice VQA, and open-ended VQA) as well as across all benchmarks (FINERS-4k, V*, and HR-Bench), as presented in Tab.2 and 3.

3. FINERS outperforms training-free methods such as [2] that utilize image patch partitioning, highlighting the effectiveness and necessity of the proposed region search mechanism.

4. The effectiveness of each proposed component is well validated through a series of ablation studies (Tab. 4).

Weaknesses:

1. The proposed FINERS framework appears to be heavily based on Seg-Zero [1], sharing a similar pipeline and loss design. As such, FINERS functions more as a great extension of Seg-Zero, integrating the region search mechanism (GSE and LPR) and supporting reasoning tasks, rather than representing a significant scientific advancement. Moreover, the region search mechanism (based on RL) has been explored in other tasks [3], which may further diminish the paper’s technical novelty and contribution.

2. Although the paper provides extensive details about the FINERS-4k dataset, such as its distribution and sample data, it does not clarify how volunteers were instructed to annotate small objects using triplet annotations (question, answer, and mask). For instance, how did the volunteers determine which object to select? How did they decide what question to ask and how to answer it? How did they ensure that no other objects also satisfied the given answer? Take Fig. 4 as an example: what if there is more than one person riding a motorcycle on the road, or more than one umbrella on the beach? This issue is particularly challenging in high-resolution images, where exhaustively checking for all possible matching objects is nearly impossible.

[2] Wang, Wenbin, et al. "Divide, conquer and combine: A training-free framework for high-resolution image perception in multimodal large language models." Proceedings of the AAAI Conference on Artificial Intelligence. Vol. 39. No. 8. 2025.

[3] Gao, Mingfei, et al. "Dynamic zoom-in network for fast object detection in large images." Proceedings of the IEEE conference on computer vision and pattern recognition. 2018.

---

> ### Author Rebuttal · Authors · 2025-07-31
>
> We sincerely thank Reviewer tyVp for the thoughtful and constructive feedback. Below, we address the main concerns in detail.
>
> ---
>
> **W1: The proposed FINERS framework appears to be heavily based on Seg-Zero [1], sharing a similar pipeline and loss design. As such, FINERS functions more as a great extension of Seg-Zero, integrating the region search mechanism (GSE and LPR) and supporting reasoning tasks, rather than representing a significant scientific advancement. Moreover, the region search mechanism (based on RL) has been explored in other tasks [3], which may further diminish the paper’s technical novelty and contribution.**
>
> Thank you for raising this important concern.
>
> While FineRS adopts the GRPO-based learning paradigm from Seg-Zero [16], it introduces several key innovations in task formulation, modular design, and training strategy.
>
> - **Task Formulation**: Unlike Seg-Zero [16] that solely addresses reason segmentation, our method focuses on **a boarder and harder task formulation** that addresses multi-output spatial reasoning, including textual answering, coarse region box, and accurate object box and points. Such unified task is conducted under a more challenging setting, ultra-small object localization in 4K UAV imagery, which significantly increases both semantic and spatial complexity.
>
> - **Architecture Design**: We propose a two-stage architecture that effectively incorporates **coarse semantic localization (GSE)** and **perception-level fine localization (LPR)**. Importantly, FINERS introduces a novel retrospective reward coupling, where LPR feedback is back-propagated to improve GSE via reinforcement learning. This closed-loop learning strategy improves both exploration efficiency and stability, and is not present in Seg-Zero or other prior works.
>
> While prior work like [3] ("Dynamic zoom-in network") also explores zoom-based strategies, their RL is applied to conventional CNN networks to navigate visual space for efficient detection. In contrast, FineRS operates in a language-conditioned setting, with region search fully guided by multi-modal reasoning in MLLMs. Moreover, [3] relies on standard detectors with fixed reward structures, whereas our method leverages instruction-driven feedback and formulates more complex multi-modal reward strategies.
>
> In summary, these differences highlight that FINERS is not a simple extension of prior works, but a task-level and framework-level advancement aimed at enabling structured multi-modal reasoning for ultra-small object understanding in high-resolution visual environments.
>
> ---
>
> **W2&Q2: Although the paper provides extensive details about the FINERS-4k dataset, such as its distribution and sample data, it does not clarify how volunteers were instructed to annotate small objects using triplet annotations (question, answer, and mask). For instance, how did the volunteers determine which object to select? How did they decide what question to ask and how to answer it? How did they ensure that no other objects also satisfied the given answer? Take Fig. 4 as an example: what if there is more than one person riding a motorcycle on the road, or more than one umbrella on the beach? This issue is particularly challenging in high-resolution images, where exhaustively checking for all possible matching objects is nearly impossible.**
>
> Thank you for this valuable comment. During dataset construction, annotators were primarily instructed to identify a single, unambiguous small object of interest from the image and compose questions that uniquely specify it (e.g., referencing color, shape, position, or context). For several cases with multiple similar objects, they were required to formulate disambiguating questions and to visually inspect the entire image to ensure no other object matched the same description.
>
> The annotation was completed by 14 volunteers, organized in pairs for mutual cross-checking. In addition, a team of 4 senior reviewers conducted a final round of quality assurance to correct ambiguities and verify consistency. This multi-stage validation process was designed to maximize precision and minimize annotation bias.
>
> While it remains difficult to guarantee absolute uniqueness in complex 4K imagery, our annotation pipeline is designed to ensure consistency and minimize overlap ambiguity. We will release detailed annotation guidelines and the full benchmark to support reproducibility and transparency.
>
> ---
>
> **Q1: Please explicitly highlight the key technical contributions of the proposed method in comparison to the Seg-Zero framework. If possible, demonstrating that the proposed modules (GSE and LPR) are also effective when applied to other frameworks or tasks would significantly strengthen the paper.**
>
> Thank you for this insightful suggestion. The contributions over Seg-Zero are discussed in **W1** above. Here, we mainly addressed the framework or task generalization.
>
> - **Module generalization**: The GSE and LPR can generalize across domains:
>
>   - **LPR**, when evaluated on the ReasonSeg dataset (See **Table R4** in Reviewer kUFY), shows promising localization performance even without any large-scale pretraining (like LISA[3]) or in-domain finetuning (like Seg-Zero[16]), demonstrating robustness to significant different domains, scales, and resolution changes.
>
>   - **GSE** also generalizes well across domains, as evidenced by QA performance on external high-resolution benchmarks. As shown in **Table 3** of main paper, its QA accuracy outperforms state-of-the-art methods by **+2.1** on V* bench, **+13.8** on HR-Bench 4k and **+17.3** on HR-Bench 8k.
>
> - **Model scalability**: We re-implement FINERS using a smaller backbone (Qwen2.5-VL-3B) and observe consistent gains over Seg-Zero 3B (see **Table R5**, Reviewer gkEp), confirming that our framework maintains effectiveness under limited capacity and is not reliant on large-scale models.
>
> These results validate the transferability and robustness of our modular design, suggesting that FineRS is broadly applicable beyond the original task.
>
> ---
>
> **Q2: Please refer to Weakness 2 and address the concerns regarding the construction of the FINERS-4k dataset. Clarifying this aspect is crucial to ensure that no potential bias was introduced during the dataset collection process.**
>
> Addressed in W2.
>
> ---
>
> Thanks again for your valuable suggestions. If you have any additional questions or would like further clarifications, we would be happy to provide more details.

---

> > ### Comment · Reviewer_tyVp · 2025-08-04
> >
> > I would like to thank the authors for addressing my concerns within the limited time of the rebuttal period. While I still view the proposed FINERS framework as a strong engineering solution that heavily builds upon Seg-Zero, as noted in my initial review, I acknowledge that the authors have effectively extended Seg-Zero to a broader set of tasks through the introduction of modules such as GSE and LPR.
> >
> > Regarding the FINERS-4k dataset, although my concerns about potential bias remain, such as the inherent challenge in ensuring diverse and unique question-answer pairs, I believe the dataset can still be valuable, provided these limitations are thoroughly discussed in the final version.
> >
> > I will maintain my overall positive assessment of the paper.

---

> ### Comment · Area_Chair_nTjB · 2025-08-03
> **Authors' Responses**
>
> Dear Reviewer,
>
> The authors have provided addition data in response to your questions. What is your view after seeing this additional information? It would be good if you could actively engage in discussions with the authors during the discussion phase ASAP, which ends on EoA (Aug 6).
>
> Best,
> AC

---

> ### Author Response · Authors · 2025-08-04
>
> We are very grateful for the reviewer's thoughtful and constructive feedback, and we are pleased that our rebuttal was helpful in addressing your concerns. We sincerely appreciate your suggestions regarding the distinction between FineRS and Seg-Zero, as well as the importance of clarifying the dataset annotation process and its potential limitations. We will add a dedicated section in the final version discussing the potential biases and limitations of FineRS-4k, including the annotation process and uniqueness verification.
>
> Thank you again for your time and for helping us strengthen our paper.

---

### Official Review · Reviewer_Du15 · 2025-07-02

**Clarity:** 2
**Significance:** 3
**Originality:** 3
**Rating:** 4
**Confidence:** 3

**Summary:**

This paper presents a reinforcement learning framework based on MLLMs for joint reasoning and fine-grained segmentation of small objects in high-resolution scenes. It introduces a new dataset with annotations that support three types of tasks including Multiple-choice Visual Question Answering (MVQA), Open-ended Visual Question Answering (OVQA), and Instruction-guided Segmentation (IS). Extensive experiments were conducted, and experimental results show that the proposed method outperforms SOTA MLLM-based approaches on the newly introduced dataset as well as other public datasets.

**Questions:**

- Each reward function in (4) and (5) contain several reward terms. Are these terms all binary like the one in (3)? Is there any need to balance the trade-off between different reward terms?
- In Table 2, the IoU metrics of some methods being compared become much worse after retraining with FINERS dataset, compared to their training-free counterparts. Any explanation for that?

**Ethical Concerns:**

["NO or VERY MINOR ethics concerns only"]

**Final Justification:**

The rebuttal clarified some of the concerns on performance sensitivity and provided additional discussion on experimental results. Overall, the FINERS-4k dataset is useful for evaluating model capability in fine-grained reasoning and segmentation of small objects. Therefore, I maintain my positive assessment of the paper.

**Limitations:**

Yes

**Paper Formatting Concerns:**

Nil

**Quality:**

3

**Strengths And Weaknesses:**

Strengths:
- It introduces a large-scale dataset of UAV imagery to address the challenging problem of reasoning and segmentation of ultra-small objects in complex high-resolution scenes.
- The newly introduced high-resolution benchmark is strategically designed to evaluate the capability of MLLMs to perform both attribute-level reasoning and pixel-level segmentation.
- It proposes a coarse-to-fine pipeline for efficient fine-tuning with reinforcement learning.

Weaknesses:
- The performance of the proposed two-stage approach heavily relies on the output quality of the first stage.
- Some details are lacking, especially the definition of reward terms in the reward functions.
- Experimental results are not discussed thoroughly enough.

---

> ### Author Rebuttal · Authors · 2025-07-31
>
> We thank the reviewer for the constructive comments on our paper. Regarding the concerns of Reviewer Du15, we provide the following responses.
>
> ---
>
> **W1: The performance of the proposed two-stage approach heavily relies on the output quality of the first stage.**
>
> Thank you for the thoughtful observation. To begin with, we should clarify that the **QA output is not dependent on the GSE’s coarse box**, as the answer is generated from the full-resolution image and instruction before region cropping. Therefore, QA accuracy remains stable even if the coarse region partially or totally misses the target.
>
> However, we acknowledge that the downstream localization performance is sensitive to the GSE output. This design reflects our coarse-to-fine search strategy inspired by human vision, where attention is first directed to a broad region and then progressively refined. Like humans, our system may fail if the initial focus region omits key information, particularly under ultra-small object conditions. To improve robustness, we introduce a retrospective reward that leverages LPR feedback to improve GSE training, which achieves an improvement of **1.1/2.5** on gIoU/cIoU (see **Table 4** in the main paper). While this mitigates some failure cases, we acknowledge the lack of a re-evaluation or correction mechanism, which is a limitation compared to human vision. In future work, we plan to explore iterative refinement or ensemble-based re-focusing strategies to enhance this pipeline.
>
> ---
>
> **W2&Q1: Some details are lacking, especially the definition of reward terms in the reward functions.**
>
> Thank you for pointing this out. We apologize for the lack of clarity. Following Seg-Zero [16], all reward terms used in Eq.4 \& Eq.5 are binary-valued.
>
> - In Eq.4:
>   1. $R_{point}=1$ if L1 distance between predicted point and GT point is less than 100 pixels.
>   2. $R_{b^{L1}}=1$ if the L1 distance between the predicted box and GT box is less than 10 pixels.
>   3. $R_{b^{IoU}}=1$ if their IoU is greater than 0.5.
>   4. $R_{response}$ is set to 1 if the predicted answer is correct (exact match for multiple choice, fuzzy match for open-ended  QA).
>   5. $R_{format}$ and $R_{think}$ are binary rewards that verify whether the output adheres to the expected JSON and reasoning formats.
>
> - In Eq.5：
>   1. $R_{{region}^{L1}}=1$ if the L1 distance between the predicted coarse box $B_r^{pre}$ and GT region $B_r^{gt}$ (see L215-221) is less than 10 pixels.
>   2. $R_{region^{IoU}}=1$ if their IoU is greater than 0.5; $R_{size}$ is 1 when the predicted coarse box is of size $512 \times 512$.
>   3. $R_{cover}$ is 1 when the ground-truth object lies fully inside the predicted region.
>
> We assign equal weights to all binary terms, following standard GRPO practices [14,16], which yielded stable performance without tuning. We will clarify these details explicitly in the final version.
>
> ---
>
> **W3&Q2: Experimental results are not discussed thoroughly enough.**
>
> Thank you for these valuable comments. We will improve the experiment discussion of both comparison and ablation studies in the final version.
>
> Regarding the observed performance drop of some baselines (e.g., LISA$^\dagger$, PixelLM$^\dagger$) after retraining, we believe this is primarily due to the significant domain shift and task complexity introduced by FineRS-4k. It consists of 4k-resolution UAV imagery with ultra-small, sparsely distributed objects, which differs drastically from the training distributions of most MLLM-based models (e.g., refcoco(+/g), ADE20k, etc.). To ensure fairness, all baselines are fine-tuned on the same FineRS-4k, starting from either LLaVA (LISA, PixelLM) or Qwen (Seg-Zero, ours). Under this low-data, high-resolution regime, fully supervised models with fixed architectures tend to overfit or underfit.
>
> A closer examination reveals different levels of robustness across baselines:
>
> - PixelLM [19] doesn't use pretrained SAM but adopts a lightweight decoder for model efficiency. However, the significant drop in training data from the original 200k images to 3626 images (8.9k samples) makes heavy LLM difficult to adapt well to this challenging task under limited training data.
>
> - LISA [3], in contrast, integrates SAM for more robust spatial grounding ability. We would also like to correct a reporting error in Table 2, where the gIoU (All) of LISA$^\dagger$ was mistakenly reported as 1.21 and should be 12.1. After this correction, LISA$^\dagger$ shows modest improvements on most metrics, demonstrating better adaptability compared to PixelLM.
>
> These observations highlight the importance of explicit spatial reasoning and modular design in adapting to high-resolution, small-object scenarios. Our FineRS adopts a coarse-to-fine reinforcement learning framework that is more data-efficient and robust to domain shifts, leading to stable performance across benchmarks and object sizes.
>
> We will provide a more in-depth analysis of these observations and additional insights into cross-method performance trade-offs in the final version.
>
> ---
>
> **Q1: Each reward function in (4) and (5) contain several reward terms. Are these terms all binary like the one in (3)? Is there any need to balance the trade-off between different reward terms?.**
>
> Addressed in W1.
>
> ---
>
> **Q2: In Table 2, the IoU metrics of some methods being compared become much worse after retraining with FINERS dataset, compared to their training-free counterparts. Any explanation for that?**
>
> Addressed in W3.
>
> ---
>
> We thank the reviewer for the helpful suggestions. If you have any further questions or suggestions, we would be happy to provide additional clarifications.

---

> > ### Comment · Reviewer_Du15 · 2025-08-07
> >
> > Thanks for the clarifications provided regarding the performance sensitivity to the GSE output and the additional discussion on experimental results. I appreciate the authors’ effort in addressing the concerns raised. After reviewing the rebuttal, I will keep my positive rating.

---

> > > ### Author Response · Authors · 2025-08-08
> > >
> > > We sincerely thank Reviewer Du15 for the positive follow-up and for acknowledging our clarifications and additional analysis. We are glad that our responses have addressed the concerns raised. In the final version, we will incorporate the discussion on GSE’s sensitivity, provide clearer explanations of the reward design, and further improve the experimental analysis to enhance the clarity and completeness of the paper.
> > >
> > > Thank you again for your time and valuable feedback throughout the review process.
> > >
> > > Warm regards,

---

> ### Comment · Area_Chair_nTjB · 2025-08-03
> **Authors' Responses**
>
> Dear Reviewer,
>
> The authors have provided addition data in response to your questions. What is your view after seeing this additional information? It would be good if you could actively engage in discussions with the authors during the discussion phase ASAP, which ends on EoA (Aug 6).
>
> Best,
> AC

---

### Official Review · Reviewer_gkEp · 2025-07-03

**Clarity:** 2
**Significance:** 3
**Originality:** 3
**Rating:** 4
**Confidence:** 4

**Summary:**

This paper proposes FINERS, a novel multi-modal large language model (MLLM) framework for precise reasoning and segmentation of ultra-small objects in high-resolution images. The key innovation is a two-stage, coarse-to-fine pipeline: the first stage predicts a coarse region and textual answer from the full image and instruction, while the second stage refines this prediction into a precise bounding box and segmentation mask. The system is trained end-to-end with reinforcement learning. The authors also introduce FINERS-4k, a new benchmark dataset specifically designed for evaluating small-object reasoning and segmentation. Experimental results demonstrate strong performance compared to state-of-the-art baselines.

**Questions:**

Questions
1.	Comparison to Agentic  Approaches:
How does FINERS compare, both in design and performance, to agent-based or active search methods like V*? What are the pros and cons of your end-to-end RL approach versus more modular or agentic frameworks?
2.	Ablation on Pretrained Models:
How does the performance of FINERS change when initialized with different pretrained MLLMs? Is the framework robust to different base model choices?

**Ethical Concerns:**

["NO or VERY MINOR ethics concerns only"]

**Final Justification:**

The rebuttal response clarified all of my prior concerns (missing baselines and the effect of the base model). Therefore, my opinion remains positive regarding this paper.

**Limitations:**

Yes

**Quality:**

2

**Strengths And Weaknesses:**

Strengths
- Significance: The task of localizing and segmenting very small objects in large, cluttered scenes is both challenging and highly relevant for real-world vision-language applications.
- Clarity: The paper is generally clear in its explanation of the methodology, and the introduction of a new dataset (FINERS-4k) is a valuable contribution to the community.

Minor Weaknesses
- Baselines: The experimental comparison is missing several important and recent baselines, especially methods focused on guided visual search for MLLMs (e.g., V*[1]). Including such comparisons would strengthen the claims.

[1] Wu, Penghao, and Saining Xie. "V?: Guided visual search as a core mechanism in multimodal llms." Proceedings of the IEEE/CVF Conference on Computer Vision and Pattern Recognition. 2024.

---

> ### Author Rebuttal · Authors · 2025-07-31
>
> We thank the reviewer for the constructive comments and positive feedback on our paper. Regarding the concerns of Reviewer gkEp, we provide the following responses.
>
> ---
>
> **W1: Baselines: The experimental comparison is missing several important and recent baselines, especially methods focused on guided visual search for MLLMs (e.g., V$*$[1]). Including such comparisons would strengthen the claims.**
>
> Thank you for pointing out the importance of guided visual search baselines. We would like to clarify that V* has been evaluated in our experiments across three benchmarks, which can be found under the SEAL[9] entry in **Table 2** and **Table 3** of the main paper. Specifically, the QA accuracy of FineRS vs. V* is 83.3/56.7 vs. 7.53/3.49 on FineRS-4k, 77.5 vs. 75.4 on V* bench, 63.8 vs 38.1 on HR-Bench 4k, and 58.1 vs 25.6 on HR-Bench 8k, respectively. We will make this connection more explicit in the final version to avoid confusion.
>
> ---
>
> **Q1: Comparison to Agentic Approaches: How does FINERS compare, both in design and performance, to agent-based or active search methods like V$*$? What are the pros and cons of your end-to-end RL approach versus more modular or agentic frameworks?**
>
> Thank you for the insightful question. Below, we compare the two paradigms in terms of design and performance.
>
> - **Design Difference**: Agentic methods like V* adopt a modular visual planning pipeline, where a controller iteratively proposes image regions and queries a frozen MLLM to gradually refine attention and predictions. These methods typically operate in a training-free or shallowly tuned setting, relying on handcrafted rules or search heuristics. In contrast, **FineRS** introduces a two-stage, coarse-to-fine framework, which contains two end-to-end trainable MLLMs (GSE and LPR). We propose a retrospective reward mechanism that uses LPR’s fine-grained feedback to enhance GSE’s localization during GRPO reinforcement fine-tuning.
>
> - **Performance Comparison**: As shown in **Table 2** and **Table 3**, FineRS consistently outperforms state-of-the-art modular methods (e.g., V*[9], MLLM-know[1], DC$^2$[10]) across all benchmarks, with gains of **+30.7/7.9** on FineRS-4k, **+2.1** on V* bench, **+13.8** on HR-Bench 4k and **+17.3** on HR-Bench 8k.
>
> In summary, unlike agentic approaches that rely on manually crafted policies, FineRS directly learns region selection and structured outputs from instruction-driven supervision, making it more adaptive and unified across tasks. However, the agentic methods are usually easier to implement, while our RL framework requires careful reward design and higher training cost. Nonetheless, the gains in spatial precision and reasoning integration justify the added complexity in high-resolution scenarios.
>
> ---
>
> **Q2: Ablation on Pretrained Models: How does the performance of FINERS change when initialized with different pretrained MLLMs? Is the framework robust to different base model choices?**
>
> Thank you for the constructive suggestion. Following previous RL-based approaches (e.g., Seg-Zero[16]), we implement our method on Qwen2.5-VL (7b) as the default backbone. Due to the limited rebuttal period, we re-implement our framework with the smaller Qwen2.5-VL (3b) model and compare it against the finetuned Seg-Zero (3b) baseline. The results in **Table R5** show that FineRS maintains promising performance even at smaller scales, highlighting its robustness to backbone size and model initialization.
>
> **Table R5: Performance comparison on the FineRS-4k test set using different pretrained MLLMs. “$\dagger$” indicates retraining on FineRS-4k.
> | Method    | S | xS| xxS | All | Color |Shape|Others|Pos.|All |
> |-------------------------------|----------------|----------------|----------------|----------------|----------------|---------------|----------------|-------------------|-------------------|
> | Seg-Zero-7B$^\dagger$ [16]     | 61.8/50.5 | 53.0/30.2 | 31.7/20.7 | 46.6/38.6 |      | --       | --       | --         | --       | --          |
> | **Ours-7B**              | **62.2/52.6** | **59.0/43.1** | **47.2/27.5** | **55.1/46.5** | 85.8/60.5 | 76.0/49.2 | 78.6/34.4 | 63.2/27.8 | 83.3/56.7 |
> | Seg-Zero-3B$^\dagger$ [16]  | **57.7**/45.3 | 47.4/22.8 | 26.3/0.86 | 41.6/29.5 | --       | --       | --         | --       | --          |
> | **Ours-3B**              | 57.4/**45.4** | **53.8/23.3** | **43.0/9.9**  | **50.4/29.9** | 65.6/50.5 | 68.6/41.0 | 60.7/21.8 | 52.6/22.2 | 65.6/47.0  |
>
> - **Note**: the columns from “s” to “All” are segmentation accuracy in gIoU/cIoU, and the columns from “color” to “All” represent QA accuracy in MVQA/OVQA.
>
> ---
>
> Thank you again for your valuable feedback. Should there be any additional concerns or suggestions, we are happy to address them.

---

> ### Comment · Area_Chair_nTjB · 2025-08-03
> **Authors' Responses**
>
> Dear Reviewer,
>
> The authors have provided addition data in response to your questions. What is your view after seeing this additional information? It would be good if you could actively engage in discussions with the authors during the discussion phase ASAP, which ends on EoA (Aug 6).
>
> Best,
> AC

---

> > ### Comment · Reviewer_gkEp · 2025-08-05
> >
> > I appreciate the reviewers' detailed clarification and additional results. I don't have further questions, and I remain positive about this paper.

---

> > > ### Author Response · Authors · 2025-08-05
> > >
> > > We are very grateful for the reviewer's positive assessment and for acknowledging our clarifications and additional results. Thank you again for your time and constructive feedback, which helped us strengthen the presentation of our work.

---

### Official Review · Reviewer_kUFY · 2025-07-05

**Clarity:** 2
**Significance:** 2
**Originality:** 2
**Rating:** 4
**Confidence:** 4

**Summary:**

The paper proposes FINERS, a two-stage multimodal large-language-model (MLLM) framework for instruction-guided reasoning and position of small objects in 4K images.

In the stage 1(GSE),  the model reads the HR image and instruction, outputs a textual answer and a fixed-size 256 × 256 coarse region.
In the stage 2(LPR), the model crops that region, predicts the position information, a precise box + two points, and calls SAM2 to obtain a pixel mask.

The stages are coupled with a locate-informed retrospective reward so that LPR’s IoU/L1 feedback reinforces GSE during vision reinforcement fine-tuning with GRPO.

The authors introduce a dataset, FINERS-4k, for evaluation.

**Questions:**

Robustness to coarse-box errors: How does FINERS detect or recover when GSE misses the target? Please quantify failure rates where the target object lies partially or fully outside the coarse region and evaluate simple fall-back strategies.

Efficiency: Please report wall-clock inference time for 4K inputs, vs VILA-HD, Seg-zero and DC^2. That would strengthen practicality.

Domain generalization: Evaluate FINERS on at least one non-UAV dataset containing small indoor objects. Comparable accuracy would indicate broader utility.

**Ethical Concerns:**

["NO or VERY MINOR ethics concerns only"]

**Final Justification:**

The authors have addressed my concerns. I increase my score

**Limitations:**

No.

**Quality:**

2

**Strengths And Weaknesses:**

[+] Coarse-to-fine RL design with retrospective reward is conceptually elegant and empirically helpful.

[+]  FINERS-4k fills a data gap for UAV, tiny-object HR tasks and could benefit the community.

[-]  Entire pipeline hinges on GSE: if the target object straddles the coarse box, both answer and mask fail.

[-]  Two stages are trained separately; no end-to-end fine-tuning —may be sub-optimal.

[-]  Hyper-parameter sensitivity (group size n, KL weight) and seed robustness are not analyzed; reproducibility risk.

---

> ### Author Rebuttal · Authors · 2025-07-31
>
> We thank the reviewer for the constructive comments on our paper. Regarding the concerns of Reviewer kUFY, we provide the following responses.
>
> ---
> **W1: Entire pipeline hinges on GSE: if the target object straddles the coarse box, both answer and mask fail.**
>
> Thank you for the thoughtful observation. At first, we would like to clarify that the QA output is not dependent on the GSE’ s coarse box, as answers are generated from the full-resolution image and instruction before region cropping. Thus, **QA accuracy remains stable** even when the coarse region fails to cover the target object.
>
> However, the downstream segmentation dose rely on GSE. This is consistent with our **coarse-to-fine design**, inspired by human vision---first identifying rough regions before refining focus. Like humans, our system may fail if the initial focus region omits key information, particularly under ultra-small object conditions. To address this, we introduce a retrospective reward that leverages LPR feedback to improve GSE training, effectively raising hit rate from 76% to 77.7% (see Table R2 below).
>
> Nevertheless, we recognize this is not a complete solution. Unlike humans, our model lacks fast re-evaluation or error correction once the region is fixed. In future work, we plan to integrate dynamic re-focusing mechanisms, such as iterative refinement or region ensemble strategies, to address this limitation more effectively.
>
> ---
>
> **W2: Two stages are trained separately; no end-to-end fine-tuning —may be sub-optimal.**
>
> Thank you for the insightful comment. Our current design adopts a two-stage training strategy to balance optimization complexity and modular flexibility. While GSE and LPR are trained separately, they are coupled via a retrospective reward during GRPO fine-tuning, enabling LPR feedback to improve GSE performance (**+1.1/2.5** in gIoU/cIoU of test set). We agree that full end-to-end optimization could enhance stage coherence, but doing so at 4k resolution presents challenges in memory and reward propagation. In future work, we aim to explore unified architectures or lightweight joint tuning to better integrate both stages.
>
> ---
>
> **W3: Hyper-parameter sensitivity (group size n, KL weight) and seed robustness are not analyzed; reproducibility risk.**
>
> In our experiments, all hyper-parameters are set to the default values in Seg-Zero [16] without specific tuning. We conduct a sensitivity analysis on key hyper-parameters, including group n, KL weight, and seeds. **Table R1** below shows that the hyper-parameter sensitivity of our method is consistent with Seg-Zero. Due to the limited rebuttal period, we present representative cases here and will include a more comprehensive analysis in the final version.
>
> **Table R1: Hyper-parameter sensitivity analysis on FineRS-4k test set.**
>
> | Hyper-params| gIoU | cIoU | MVQA| OVQA|
> |------------------|------|------|------|------|
> | Group 8  | 55.1 | 46.5 | 83.3 | 56.7 |
> | Group 6    | 54.3 | 43.7 | 82.6 | 53.5 |
> | KL 5e-3    | 55.1 | 46.5 | 83.3 | 56.7 |
> | KL 5e-2    | 54.4 | 46.1 | 82.9 |  55.8|
> | Seed 48    |  53.9 | 45.4 | 82.0 | 56.2|
>
> ---
>
> **Q1: Robustness to coarse-box errors: How does FINERS detect or recover when GSE misses the target? Please quantify failure rates where the target object lies partially or fully outside the coarse region and evaluate simple fall-back strategies.**
>
> As clarified in W1, **QA outputs are not dependent on GSE**. However, **segmentation performance is sensitive to coarse-box accuracy**. When GSE misses the target, LPR segmentation degrades. To quantify this, we report a **22.3% failure rate** where the GT mask lies partially or fully outside the GSE region. Even so, FINERS still outperforms Seg-Zero by **+10.5 gIoU / +6.9 cIoU**. Besides, we also list two baselines: “one-stage” (can be seen as Seg-Zero with QA) and “w/o retrospective reward”, which further demonstrate the effectiveness of our two-stage design.
>
> Besides, we implement a simple fallback strategy: expanding GSE’s output box to $800\times800$ pixels, which boosts coarse-box hit rate from 77.7% to 82.6%. However, the altered crop size introduces more context to LPR, slightly compromising final performance. In Table R2, we additionally report **upper-bound performance** using GT-based coarse boxes. These results confirm that our **two-stage design with retrospective reward** is effective, and we will explore more robust fallback mechanisms in future versions.
>
> **Table R2: Analysis on failure rate mitigation strategies on FineRS-4k test set.**
>
> | Method | gIoU  | cIoU  | Box Acc (%) |
> |-------------------------------|-------|-------|------------|
> | FineRS (GSE + LPR)            | 55.1  | 46.5  | 77.7      |
> | Seg-Zero$^\dagger$            | 44.6  | 38.6  | 77.7     |
> | One-stage                   | 42.3  | 41.8  | --       |
> | w/o Retrospective reward  | 54.0  | 44.0  | 76.0     |
> | GSE Box Expand 800×800        | 54.5  | 46.2  | 82.6   |
> | GT Coarse Box + LPR      | 70.6 | 68.2 | 100  |
>
> ---
>
> **Q2: Efficiency: Please report wall-clock inference time for 4K inputs, vs VILA-HD, Seg-zero and DC$^2$. That would strengthen practicality.**
>
> Thanks for the insightful comment. We report the average wall-clock inference time for 4K-resolution inputs in **Table R3**. All models were evaluated on a single A100 GPU with consistent runtime environments. As shown, compared to SEAL[9] and DC2[10], our method and Seg-Zero[16] exhibit higher inference latency due to the use of CoT reasoning. Despite our two-stage framework requires extra inference time, this design is essential for achieving precise reasoning and segmentation of ultra-small objects in high-resolution scenes. Notably, the VILA-HD[28]'s results in Tab.3 are copied from its technical report, since its inference code is released after our submission. Due to the limited rebuttal time, we fail to implement their testing code and don't report its time here. We promise to complete this experiment and update the complete results in the next version.
>
> **Table R2: Inference latency and performance of different methods on FineRS-4k test set (left two columns) and HR-bench-4k (right two columns).**
>
> | Method  | MVQA/OVQA/gIoU/cIoU   | Time (s/img) | QA ACC.  | Time (s/img)  |
> |-----------------------------|-----------------------------------------|---------------------------|-----------------------|-------------------|
> | SEAL-7b [9]  | 7.53 / 3.49 /  ----  / ----   | 1.21   | 38.1   | 1.15 |
> | Seg-Zero$^\dagger$-7b [16]  | ---- / ---- / 46.6 / 38.6   | 8.67   | --  | --   |
> | DC$^2$-7b [10]    | 39.2 / 17.8 / ---- / ----    | 2.69   | 50.0    | 2.90   |
> | **Ours-7b**   | **83.3 / 56.7** / **55.1 / 46.5**  | 7.31 (GSE) + 5.57 (LPR+SAM2) | **63.8**  | 6.35 (GSE only) |
>
> ---
>
> **Q3: Domain generalization: Evaluate FINERS on at least one non-UAV dataset containing small indoor objects. Comparable accuracy would indicate broader utility.**
>
> Thank you for your valuable suggestion. We have carefully reviewed the existing referring segmentation datasets but did not find an exact dataset specifically designed for non-UAV, high-resolution, and small indoor objects. To address this, we conduct **zero-shot evaluations** on ReasonSeg (proposed by LISA [3]) test set, which includes **ground-level daily scenes with moderate-to-large object sizes**, significantly different from **aerial, ultra-high-resolution imagery and ultra-small object** focus of FineRS-4k.
>
> We categorize test samples by mask-to-image area ratio into Large ($>$50%), Middle (10–50%), and Small ($<$10%). Notably, **ReasonSeg's "Small" objects are still much larger than FineRS-4k’s (<1\%), introducing a challenging domain and scale gap.** Despite this, our two-stage model achieves better gIoU on Small objects, outperforming finetuned baselines like LISA$^\dagger$ and Seg-Zero$^\dagger$. This confirms the effectiveness of our coarse-to-fine strategy in segmenting small targets, even under mismatched resolution and context. Moreover, we find that our LPR shows strong adaptability across all sizes and outperforms other methods even without domain-specific finetuning, highlighting its robustness.
>
> These results indicate that FINERS, while designed for ultra-small aerial targets, retains strong generalization capability to conventional domains, especially when object granularity matters. We will include these findings in the final version.
>
> **Table R4: Performance on the ReasonSeg test set (IoU: gIoU / cIoU). Results are grouped by object size.**
>
> | Method    | Large (316 samples) | Middle (388 samples) | Small (72 samples) | All  |
> |------------------------------|---------------------|-----------------------|--------------------|-------------|
> | FineRS 7B (resize)   | 25.2 / 4.92         | 41.4 / 15.6           | **42.3** / 7.12        | 35.0 / 7.16 |
> | FineRS 7B (padding)    | 27.5 / 5.02         | 43.2 / 16.1           | **40.0** / 9.42        | 36.1 / 7.52 |
> | LPR only    | 59.0 / 52.4         | **54.4** / **35.3**           | **45.7** / **24.6**    | 56.6 / 42.1 |
> | Seg-Zero† 7B [16]  | 49.3 / 41.7         | 46.1 / 30.9           | 35.8 / 15.4        | 47.1 / 38.8 |
> | Seg-Zero 7B [16]  | 65.3 / 55.2         | 53.5 / 31.4           | 39.0 / 16.0        | 57.5 / 52.0 |
> | LISA† 7B [3]   | 0.44 / 0.34         | 4.08 / 1.38           | 8.14 / 1.12        | 2.98 / 0.55 |
> | LISA 7B [3]    | 55.3 / 56.7         | 34.2 / 26.9           | 20.3 / 24.8        | 48.7 / 48.8 |
>
> - *Note:* “†” denotes the corresponding methods are finetuned with FineRS-4k without pretraining on large-scale referring segmentation datasets.
> - “resize” means we directly resize ReasonSeg image to meet our model’s require ($1920\times1080$). “padding” means that the low-resolution image are padded to meet our model’s resolution.
>
> ---
>
> We hope these clarifications address your concerns. We are grateful for your thoughtful feedback and are committed to improving our work in future versions.

---

> ### Comment · Area_Chair_nTjB · 2025-08-03
> **Authors' Responses**
>
> Dear Reviewer,
>
> The authors have provided addition data in response to your questions. What is your view after seeing this additional information? It would be good if you could actively engage in discussions with the authors during the discussion phase ASAP, which ends on EoA (Aug 6).
>
> Best,
> AC

---

> ### Author Response · Authors · 2025-08-05
>
> Dear Reviewer kUFY,
>
> Thank you again for your valuable feedback. We would like to kindly confirm whether all your concerns have been adequately addressed. If there is anything further we can clarify, we are happy to provide additional explanations.
>
> Best regards,
>
> Authors of #1107

---

> > ### Author Response · Authors · 2025-08-09
> >
> > Dear Reviewer kUFY,
> >
> > As the discussion period will conclude in just a few hours, we are writing to follow up on our rebuttal. We saw that you have acknowledged receipt of our response, and we thank you for your engagement.
> >
> > We sincerely hope that our clarifications and additional results have been helpful in addressing your initial concerns. We feel these additions have substantially strengthened our manuscript, and we hope this improved version can earn your support.
> >
> > Thank you again for your time and continued consideration of our work.
> >
> > Sincerely, Authors of 1107

---

### Comment · Area_Chair_nTjB · 2025-08-02
**Discussion with Authors**

Dear Reviewers,

The discussion period with the authors has now started. It will last until Aug 6th AoE. The authors have provided responses to your questions. I request that you please read the authors' responses, acknowledge that you have read them and start discussions with the authors RIGHT AWAY if you have further questions, to ensure that the authors enough time to respond to you during the discussion period.

Best,
AC

---

### Note · Authors · 2025-08-12

Thank you for this final opportunity. We express our sincere gratitude to all the reviewers for their valuable insights and constructive engagement throughout the review process! We thank all reviewers for their recognition of our work’s strengths on **an elegant and effective coarse-to-fine RL framework** (*Reviewer kUFY, gkEp, Du15, tvVp*), **Task significance**(*Reviewer kUFY, gkEp, Du15, tvVp*), **Dataset contribution for communities** (*Reviewer kUFY, gkEp, Du15, tvVp*), and stronger performance over baselines (*Reviewer gkEp, Du15, tvVp*).

In the initial review, a major concern raised by *Reviewer kUFY* and *Reviewer Du15* is the **performance sensitivity of GSE module**. In response, we have analyzed GSE’s hit rate and its upper-bound, quantified the improvement from the retrospective reward, explored simple fallback strategies, and outlined future plans for more robust region refinement.

Another concerns is the **contributions over baseline SegZero[16]**. We have clarified our innovations in **task formulation, modular design and training strategy**. We strength that we target a broader and more challenging task, and design a tightly coupled coarse-to-fine framework with a reinforcement learning strategy that improves both training efficiency and scalability.

We are trying to address the concerns and polish our paper in the revised version to make our work better understandable to the wider community. Specifically, we will:

1.	Update results on dataset/backbone generalization, hyperparameter ablations, GSE sensitivity, and efficiency.
2.	Provide more detailed discussion with SegZero[16] and V*[9], enhance experimental discussion, and expand the reward design explanation.
3.	Add a section discussing the annotation guidance and potential bias mitigation.

Finally, we are greatly encouraged by the reviewers’ recognition of our contribution to the community. We want to emphasize that our work contributes to the community not only by **a new HR benchmark for ultra-small object reasoning and segmentation, but also by proposing an effective coarse-to-fine RL framework with demonstrated promising performance**. We will release the data, code, and models publicly, and **believe that our work is worth publishing to stimulate further discussion**.

Best regards,
Authors of 1107

---

### Decision · Program_Chairs · 2025-09-17

**Decision:**

Accept (poster)

**Comment:**

This paper proposes a multimodal LLM with an RL-based reasoning approach for understanding and localization of very small objects in high resolution images, along with a new benchmark dataset for the task. Four reviewers provided final scores of 4 x borderline accept. The reviewers appreciated the work for its novel proposed approach, the significance of its task and its dataset. The reviewers' initial concerns were adequately addressed during the author-reviewer discussion phase and they all rated the work positively post-discussions. The AC concurs with the reviewers' consensus and recommends acceptance. Congratulations! The authors should make the changes that they have promised in the camera ready version of their paper.